# Dendritic sodium spikes are required for long-term potentiation at distal synapses on hippocampal pyramidal neurons

Yujin Kim[1,2†], Ching-Lung Hsu[1,2†], Mark S Cembrowski[1], Brett D Mensh[1], Nelson Spruston[1,2]*

[1]Janelia Research Campus, Howard Hughes Medical Institute, Ashburn, United States; [2]Department of Neurobiology, Northwestern University, Evanston, United States

**Abstract** Dendritic integration of synaptic inputs mediates rapid neural computation as well as longer-lasting plasticity. Several channel types can mediate dendritically initiated spikes (dSpikes), which may impact information processing and storage across multiple timescales; however, the roles of different channels in the rapid vs long-term effects of dSpikes are unknown. We show here that dSpikes mediated by $Na_v$ channels (blocked by a low concentration of TTX) are required for long-term potentiation (LTP) in the distal apical dendrites of hippocampal pyramidal neurons. Furthermore, imaging, simulations, and buffering experiments all support a model whereby fast $Na_v$ channel-mediated dSpikes (Na-dSpikes) contribute to LTP induction by promoting large, transient, localized increases in intracellular calcium concentration near the calcium-conducting pores of NMDAR and L-type $Ca_v$ channels. Thus, in addition to contributing to rapid neural processing, Na-dSpikes are likely to contribute to memory formation via their role in long-lasting synaptic plasticity.

*For correspondence: sprustonn@janelia.hhmi.org

†These authors contributed equally to this work

Competing interests: The authors declare that no competing interests exist.

## Introduction

Hebbian remodeling of neural circuits, encapsulated by the phrase 'neurons that fire together wire together', is the leading candidate mechanism for learning in the brain (*Shatz, 1992*; *Cooper, 2005*). Thus, the finding that synaptic plasticity often has properties compatible with Hebbian learning has been a powerful driver of experimental and theoretical studies of changes in synaptic weights. The simple idea is that coincident presynaptic and postsynaptic activity results in changes in synaptic strength such as long-term potentiation (LTP). Many in vitro studies support a model in which modifiable synapses detect presynaptic activity through glutamate binding to postsynaptic receptors and detect postsynaptic activity through depolarization, which relieves magnesium block of *N*-methyl-D-aspartate (NMDA) receptors (NMDARs) and activates voltage-gated calcium ($Ca_v$) channels. Both of these effects mediate postsynaptic calcium entry, which is believed to be a critical activator of the biochemical steps necessary for increases in synaptic strength (*Madison et al., 1991*; *Bliss and Collingridge, 1993*; *Blackstone and Sheng, 2002*; *Cavazzini et al., 2005*). Four decades of research have led to this general picture, but important questions about the events leading to the critical calcium entry remain unanswered.

Three distinct ideas have been considered for the type of postsynaptic depolarization required to produce the calcium entry leading to Hebbian LTP (*Williams et al., 2007*):

1. Postsynaptic axo-somatic action potential firing is necessary, and this signal reaches synapses in the form of backpropagating action potentials (bAPs),
2. Postsynaptic axo-somatic action potential firing is not necessary; rather, localized passive synaptic depolarization is sufficient,
3. Postsynaptic axo-somatic action potential firing is not necessary and passive synaptic depolarization is not sufficient; rather, localized synaptic depolarization must activate dendritic nonlinearities, such as locally initiated dendritic spikes (dSpikes).

**eLife digest** When we explore somewhere new, we activate a region of the brain that processes spatial information called the entorhinal cortex. This brain region stimulates the brain's memory-formation center, known as the hippocampus, which in turn forms a spatial memory of the new place. The process of forming these memories involves strengthening nerve connections, including those between the entorhinal cortex and the hippocampus.

Groups of neurons that produce synchronized electrical activity will naturally strengthen the nerve connections between them. This led scientists to predict that synchronized electrical activity between neurons in the entorhinal cortex and the hippocampus may contribute to the formation of spatial memories.

Previous research revealed that hippocampal neurons produced short bursts of electrical activity that are localized at specific sites along their branched nerve processes that extend out of the cell body and are where inputs from other neurons are received. These types of localized electrical activity have been associated with a strengthening of the nerve connections between the entorhinal cortex and the hippocampal neurons. Ion channels that allow calcium to flow through these neurons' cell membranes had been identified as a potential source of these local electrical activities, and calcium is responsible for the strengthening of nerve connections. But it remained unclear whether channels that allow only sodium ions to flow through might also be involved.

Kim, Hsu et al. have now investigated this question by devising a way to selectively block the electrical activity produced by sodium ion channels on the branched nerve processes of hippocampal neurons. Slices of rat brain were collected and an inhibitor that specifically affected the sodium channels was delivered to the brain slices. Electrodes were used to stimulate the inputs from the entorhinal cortex, and to monitor the resulting electrical activity in the hippocampal neurons.

Kim, Hsu et al. analyzed the results and reproduced them using computer simulations, which showed that sodium ion channels are essential for triggering brief electrical events within the individual branches of nerve processes. These local electrical events appeared to activate calcium channels to produce highly concentrated, short-lived calcium signals that are necessary for strengthening nerve connections. Future studies will determine whether local electrical activity mediated by sodium channels is also involved in strengthening nerve connections between other types of neurons, and how this mechanism affects the formation of memories.

At any given synapse at any moment in time, these three mechanisms of postsynaptic activity detection are mutually exclusive. However, all three mechanisms may exist in the brain, for example, in different cell types or even within the same cell at different synapses, different stages of development, or possibly in response to different activation patterns at the same synapse. Such heterogeneity would be consistent with the idea that LTP is not a single phenomenon, but rather a collection of mechanisms that can produce a long-lasting increase in synaptic strength (*Malenka and Bear, 2004*).

In each of the three cases above, the concerted action of multiple synapses is required—the so-called 'cooperativity' requirement for LTP (*Feldman, 2012*)—but the patterns of synaptic activation that result in LTP are quite different. In the first case, the synapses that drive action potential firing could be located anywhere in the dendritic tree and plasticity would occur at all synapses that experience significant depolarization as a result of the bAP. This is the most conventional interpretation of Hebbian LTP. In the second and third conditions, synapses have to be co-localized in the dendrites in order to depolarize each other sufficiently to produce Hebbian LTP. In these conditions, the postsynaptic axon does not have to 'fire' at all; the third condition additionally requires that the co-localized synapses produce enough depolarization to trigger a dSpike, which may occur with or without axonal firing. Although these conditions may still be classified as 'Hebbian LTP', because of the requirement for co-incident presynaptic and postsynaptic activation, the existence of LTP under these conditions would imply that Hebbian-like LTP occurs at finer spatial scales than the more cell-wide form that most current models employ. Thus, during behavior, neurons that fire may undergo LTP, but even a neuron that is synaptically activated but axonally silent during a behavioral event can be recruited to participate in the neural engram.

Interestingly, hippocampal place cells behave in a manner suggestive of a form of Hebbian LTP that may not require postsynaptic action potential firing: many cells are silent when the animal first goes to a new place, and then they are recruited to participate in the network representation of the spatial map (*Frank et al., 2004*). This suggests that if synaptic plasticity contributes to reshaping the network upon initial exposure to a new environment, it need not be conventional Hebbian plasticity, but rather a modified form of Hebbian plasticity that does not require axonal firing, such as the second and third conditions described above. Consistent with this idea, hippocampal neurons receive spatially tuned synaptic inputs even when they are silent (*Lee et al., 2012*). Understanding whether and how such inputs can drive synaptic plasticity will be an important step toward understanding how spatial memories are formed in the hippocampus.

If axonal action potential firing is required for synaptic plasticity, memories can only be stored in active neurons. On the other hand, if it is not required, memories can *also* be formed in silent neurons. Furthermore, plasticity that is induced by dSpikes that remain localized to individual dendritic branches has been proposed to enhance the memory-storing capacity of individual neurons (*Poirazi and Mel, 2001*; *Mehta, 2004*; *Wu and Mel, 2009*). Collectively, these considerations underscore the importance of understanding the dendritic events leading to the postsynaptic calcium entry necessary for the induction of LTP.

At synapses from the perforant path (PP; which carries predominantly spatial information from the entorhinal cortex) to the distal apical tuft of hippocampal CA1 pyramidal neurons, LTP requires strong synaptic activation, and LTP induction can have a significant impact on the output of CA1 neurons (*Colbert and Levy, 1993*; *Remondes and Schuman, 2002*; *Ahmed and Siegelbaum, 2009*; *Takahashi and Magee, 2009*). In previous work, we showed that LTP at these synapses does not require bAPs; rather, LTP is correlated with the initiation of dSpikes, which often do not trigger action potential firing and bAPs (*Golding and Spruston, 1998*; *Golding et al., 2002*). This pathway therefore offers an ideal opportunity to study the potential role of dSpikes in the induction of LTP.

The hypothesis that dSpikes are a causal step in the induction of LTP has not been directly tested, owing to the difficulty of blocking them selectively. Three types of dSpikes have been described, which have been named according to the primary channel type supporting the regenerative event: dendritic sodium spikes (Na-dSpikes) and dendritic calcium spikes (Ca-dSpikes) are mediated primarily by voltage-gated sodium ($Na_v$) channels and $Ca_v$ channels, respectively, while dendritic NMDA spikes (NMDA-dSpikes) are mediated primarily by NMDAR channels (*Schwartzkroin and Slawsky, 1977*; *Andreasen and Lambert, 1995*; *Schiller et al., 1997*; *Golding et al., 1999*; *Larkum et al., 1999*; *Spruston, 2008*; *Larkum et al., 2009*; *Major et al., 2013*). NMDAR and $Ca_v$ channels are known to contribute to the induction of LTP at PP-CA1 synapses (*Golding et al., 2002*; *Takahashi and Magee, 2009*), but it is difficult to disentangle the importance of the calcium permeability of these channels from their roles in mediating regenerative dendritic voltage changes. Furthermore, the importance of dendritic $Na_v$ channels and Na-dSpikes has not been addressed, mostly because these channels are essential for action potential firing in presynaptic axons and terminals, thus making it difficult to block them without inhibiting synaptic transmission.

One strategy to block postsynaptic $Na_v$ channels without affecting presynaptic $Na_v$ channels is to use the intracellular blocker QX-314; however, this drug also blocks $Ca_v$ channels, voltage-gated potassium ($K_v$) channels and has effects on other channels and receptors, making the overall consequences difficult to interpret (*Nathan et al., 1990*; *Andrade, 1991*; *Oda et al., 1992*; *Lambert and Wilson, 1993*; *Martin et al., 1993*; *Otis et al., 1993*; *Perkins and Wong, 1995*; *Talbot and Sayer, 1996*). As an alternative strategy, we used a relatively low concentration of bath-applied tetrodotoxin (TTX; 20 nM) to achieve partial block of the $Na_v$ channels (*Kaneda et al., 1989*; *Madeja, 2000*), which we demonstrate is able to inhibit postsynaptic dSpikes mediated by $Na_v$ channels without blocking presynaptic action potential firing or synaptic transmission. This strategy works because the density of $Na_v$ channels is much lower in dendrites than in axons. As a result, the safety factor for spike initiation is lower in dendrites, so partial block of $Na_v$ channels affects postsynaptic (dendritic) dSpikes much more than presynaptic (axonal) action potentials (*Mackenzie and Murphy, 1998*). Using this strategy, we demonstrate that Na-dSpikes are necessary for the induction of LTP in response to theta-burst stimulation (TBS) of the PP synapses in the apical tuft dendrites of CA1 pyramidal neurons, and we offer an explanation for why these spikes are an essential mechanism.

# Results

To determine whether a low concentration of TTX (20 nM) could be used without interfering with synaptic transmission over a range of stimulus intensities, we performed whole-cell recordings from CA1 pyramidal neurons in rat hippocampal slices while stimulating the PP ('Materials and methods') to activate synapses in the distal apical tuft dendrites (henceforth 'PP → CA1$_{tuft}$' synapses). We adjusted stimulus intensity to yield single PP → CA1$_{tuft}$ excitatory postsynaptic potentials (EPSPs) of 3–10 mV at the soma, which likely corresponds to activation of several tens of synapses onto the apical tuft dendrites of the recorded neuron (*Golding et al., 2005*; *Nicholson et al., 2006*). We chose 20 nM TTX (henceforth 'low TTX') because it was the highest concentration we could use without obviously reducing the size of EPSPs. The IC$_{50}$ of TTX on Na$_v$ channels has been estimated as 6–10 nM in dissociated hippocampal neurons (*Kaneda et al., 1989*; *Madeja, 2000*), but it may be higher for slice experiments due to limited drug penetration in brain slices. Single EPSPs in response to low-frequency (single shock) stimulation were not affected by bath application of low TTX (control: 4.57 ± 0.36 mV, low TTX: 5.02 ± 0.49 mV, n = 7; p > 0.05 by Student's t-test; data not shown). We also tested the effects of low TTX on the responses to low-frequency or high-frequency (5 stimuli at 100 Hz) activation of the PP, while using localized application of a higher concentration of TTX (10 μM in the application pipette) to the vicinity of the soma and proximal axons ('Materials and methods'; *Figure 1A*) in order to prevent somatic or axonal action potential initiation. Bath application of low TTX had no significant effect on PP → CA1$_{tuft}$ EPSPs in response to either type of stimulation, across a wide range of stimulus intensities resulting in EPSPs from ~3 mV (single shock) up to ~27 mV (high-frequency burst; *Figure 1B–G*; *Figure 1—source data 1*). These results suggest that excitatory synaptic transmission is not affected by bath application of low TTX within the range of stimulus intensities and frequencies tested here.

Consistent with this result and with the notion of a high safety factor for action potential initiation in the axon (*Coombs et al., 1957*; *Mainen et al., 1995*; *Raastad and Shepherd, 2003*; *Clark et al., 2005*; *Khaliq and Raman, 2006*; *Kole and Stuart, 2008*; *Hu and Jonas, 2014*), low TTX did not block action potential firing in response to somatic current injection, though it did raise the voltage threshold and reduce action potential amplitude slightly, as expected from a reduction of Na$_v$ channel availability (*Figure 1—figure supplement 1A–C*; *Figure 1—source data 2*; see also *Mackenzie and Murphy, 1998*; *Magee and Carruth, 1999*; *Fan et al., 2005*). In addition, we tested the effects of low TTX on action potentials evoked by repeated bursts of antidromic stimulation of CA1 axons and found that low TTX only affected antidromic action potential firing at low stimulus intensities. At high stimulus intensities, antidromic spikes were not blocked by low TTX (*Figure 1—figure supplement 1D,E*; *Figure 1—source data 2*). Together with the lack of effect of low TTX on synaptic responses, these data suggest that the effects of low TTX on presynaptic action potential firing and glutamate release are minimal.

To analyze the effects of low TTX on dSpike initiation, we performed recordings from the primary apical dendrite (n = 9; 200–320 μm from the soma, mean distance = 250 μm; about 50–80% of the distance from the soma to the end of the apical tuft). Dendritic recording is necessary to increase the probability of detecting dSpikes originating in the apical tuft, because their steep attenuation makes them impossible to detect hundreds of microns away in somatic recordings. We used a stimulus pattern known to cause dSpikes (*Golding et al., 2002*), which consisted of PP stimulation in high-frequency bursts (5 stimuli at 100 Hz), repeated five times at theta frequency (5 Hz; a 'theta-burst stimulation', TBS, consisting of 25 stimuli in total). Stimulus intensity was set to evoke single PP → CA1$_{tuft}$ EPSPs of 4–10 mV recorded in the dendrites, approximately the same as that required to produce single PP → CA1$_{tuft}$ EPSPs of 2–5 mV recorded at the soma (*Golding et al., 2005*). This stimulus pattern is intended to mimic the theta rhythm observed both in the hippocampus and in the entorhinal cortical neurons forming the PP (*Buzsáki and Moser, 2013*), and is known to be an effective stimulus for induction of LTP in the hippocampus (*Larson et al., 1986*; *Phillips et al., 2008*). A TBS is normally repeated multiple times to induce LTP, but in this first series of experiments, we used a single TBS in the presence of low TTX followed by another single TBS after washout of TTX. We used this order of drug application and avoided repeating TBS in each condition in order to prevent the induction of LTP, which would confound the comparison of responses in the presence and absence of TTX.

Consistent with previous results (*Golding and Spruston, 1998*; *Golding et al., 2002*; *Losonczy and Magee, 2006*), we observed three types of regenerative dendritic responses: (1) bAPs, (2) large dSpikes, and (3) small dSpikes (also known as 'spikelets'). These three kinds of events were

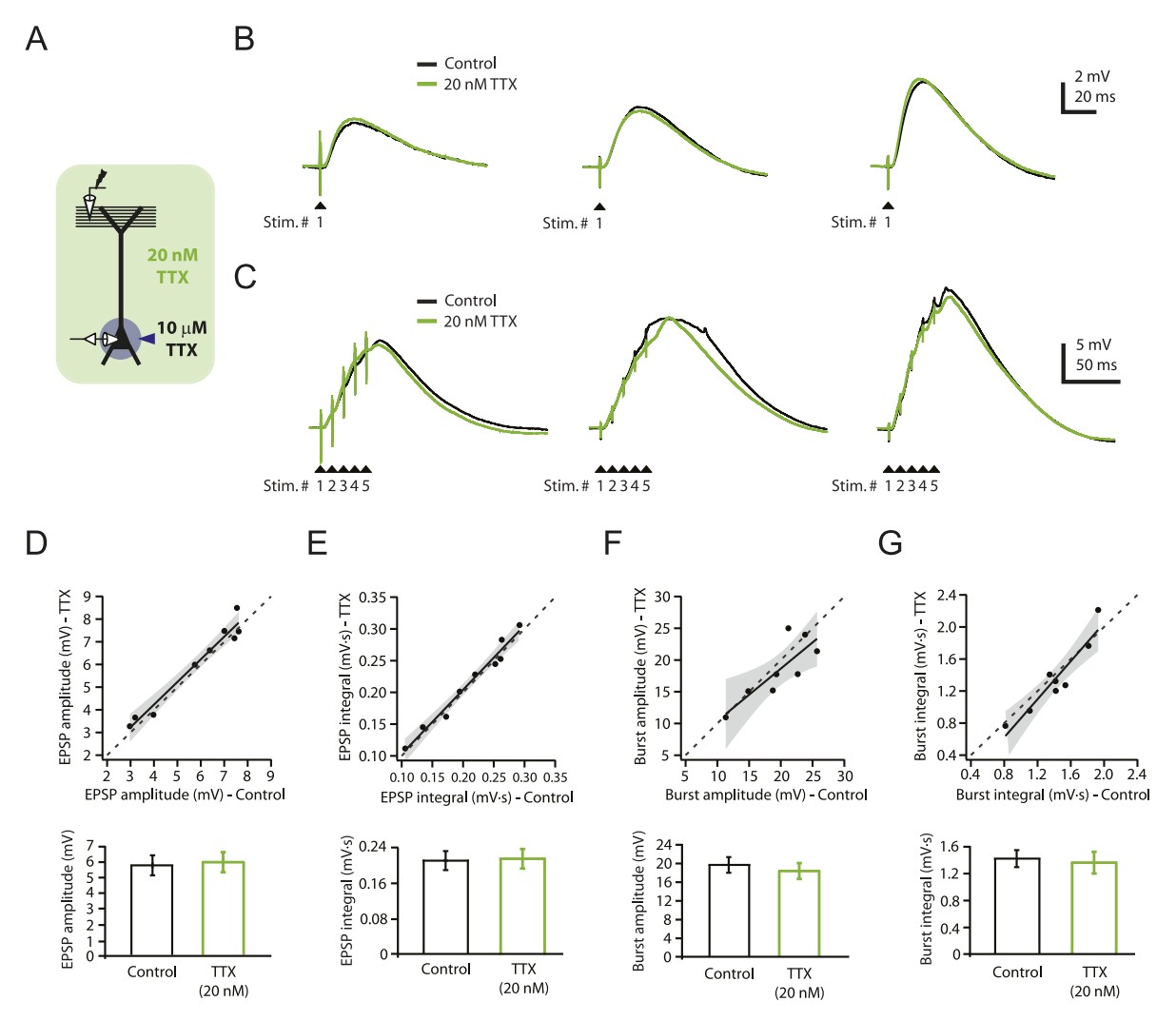

**Figure 1**. Reducing Na$_v$ channel availability with 20 nM TTX does not affect synaptic transmission at PP → CA1$_{tuft}$ synapses. (**A**) Experimental configuration showing somatic whole-cell recording with presynaptic stimulation of the perforant pathway (PP), 10 µM TTX locally applied to the soma, and bath application of 20 nM TTX. (**B**, **C**) Representative traces of somatically recorded voltage in response to single-shock stimulation (**B**) or high-frequency burst stimulation (5 stimuli at 100 Hz; **C**) in control and 20 nM TTX. Traces are from three different cells. (**D**–**G**) Summary of effects of 20 nM TTX on EPSP amplitude and integral (**D**, **E**, single-shock, n = 9; **F**, **G**, burst, n = 8). *Top*. Scatter plots of the amplitude or integral of responses in 20 nM TTX vs control. Each point represents data from one cell. Solid lines represent a linear fit to data points, with shaded areas representing the 95% confidence band of the fit. Dashed lines are the unity line. *Bottom*. Bar graphs of the amplitude or integral of responses in control and 20 nM TTX.

The following source data and figure supplement are available for figure 1:

**Source data 1**. Source data for *Figure 1*.

**Source data 2**. Source data for *Figure 1—figure supplement 1*.

**Figure supplement 1**. Effects of reducing Na$_v$ channel availability with 20 nM TTX on somatic action potentials.

distinguishable in most cases by their voltage amplitude, first temporal derivative of voltage (dV/dt) and onset kinetics (*Figure 2A,B*; *Figure 2—source data 1*; 'Materials and methods'). Large dSpikes reflect dSpikes that have propagated actively from their initiation site to the recording site. However, dSpikes can fall below threshold for active propagation (often because of large impedance drops at branch points) and then propagate passively to the recording electrode, where they appear as small

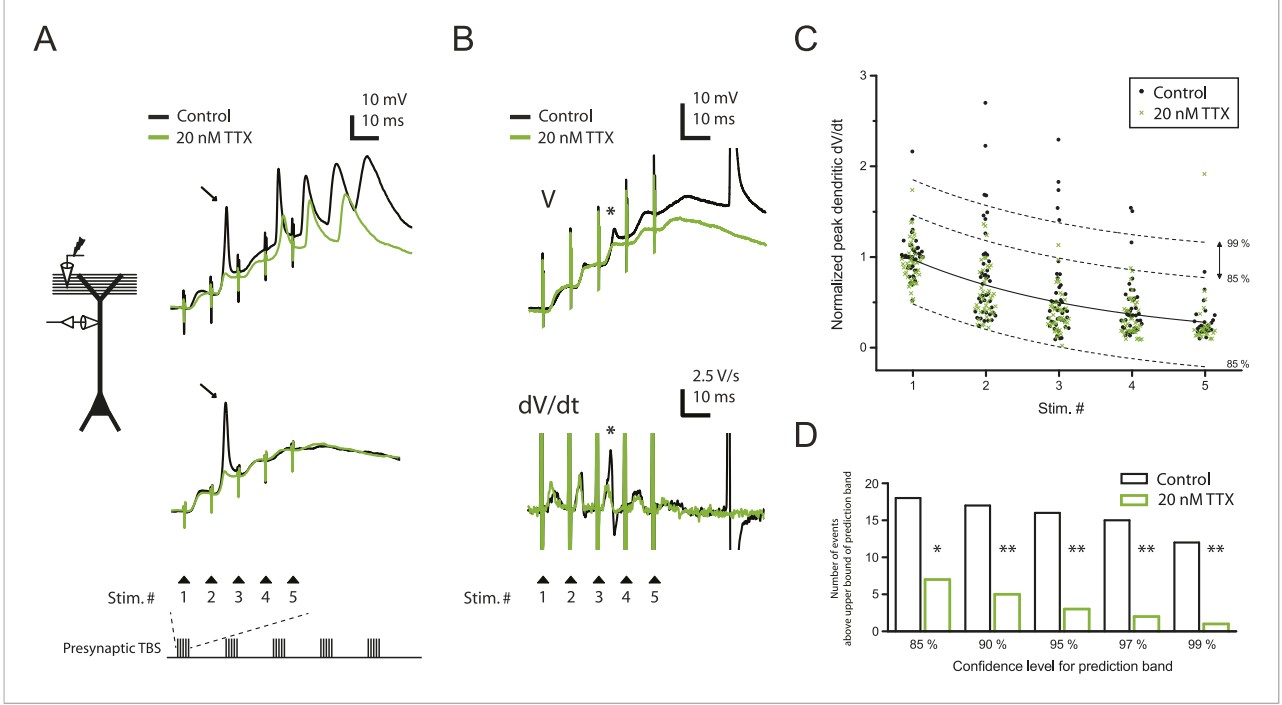

**Figure 2**. Reducing Na$_v$ channel availability with 20 nM TTX inhibits postsynaptic dSpikes evoked by presynaptic TBS of PP → CA1$_{tuft}$ synapses. (**A**) Left, experimental configuration showing dendritic whole-cell recording with presynaptic stimulation of the PP. Right, traces of dendritically recorded voltage in response to a single burst of presynaptic theta-burst stimulation (TBS) from two different neurons with large dSpikes in control (arrows; see *Figure 2—figure supplement 2*) and the corresponding responses in 20 nM TTX. In the *top* traces, the large-amplitude dendritic events following the fourth and fifth stimulus in the burst are likely to be bAPs ('Materials and methods'). Recordings were performed at 280 and 320 µm from the soma for the *top* and the *bottom* traces, respectively. (**B**) Representative traces of dendritically recorded voltage (V; *top*) and the first temporal derivative of voltage (dV/dt; *bottom*) in response to a single burst of presynaptic TBS from a neuron with a small dSpike (spikelet; asterisk) in control and the corresponding responses in 20 nM TTX. Experimental configuration is as shown in the inset in **A**. Recording was performed at 290 µm from the soma (note: a large-amplitude dendritic event following the fifth stimulus in the burst is truncated. It is likely to be a bAP; see 'Materials and methods'). (**C**) Summary of the dendritic recordings, showing peak first temporal derivative of dendritically recorded voltage (dV/dt), normalized to the median value for Stim. # 1 of the five bursts (in control) from the given cell, plotted as a function of Stim. # in each burst in control and 20 nM TTX (n = 9 cells; positions with a large-amplitude dendritic event in the control condition were excluded; see 'Materials and methods'). The solid line represents an exponential fit to the whole data set (control and 20 nM TTX), and the dashed lines represent the prediction band at the 85% confidence level and the upper bound at the 99% confidence level. (**D**) Summary of effects of 20 nM TTX on small dSpikes (spikelets), showing the number of events above the upper bound of prediction band at different confidence levels (i.e., small dSpikes; see 'Materials and methods') in control and 20 nM TTX. **p < 0.01, *p < 0.05 by binomial test.

The following source data and figure supplements are available for figure 2:

**Source data 1**. Source data of *Figure 2*.

**Source data 2**. Source data of *Figure 2—figure supplement 1–4*.

**Figure supplement 1**. Peak amplitude and the first temporal derivative of dendritically recorded voltage in response to TBS plotted as a function of stimulus position in TBS.

**Figure supplement 2**. Distinguishing between bAPs and large dSpikes in dendritic recordings.

**Figure supplement 3**. Voltage and the first temporal derivative of voltage of the clearest small dSpikes (spikelets).

**Figure supplement 4**. Small dSpikes (spikelets) identified by a second method.

dSpikes (spikelets) (*Golding and Spruston, 1998*; *Golding et al., 2002*; *Jarsky et al., 2005*; *Losonczy and Magee, 2006*; *Katz et al., 2009*). dSpikes that are generated far from the recording site may not be visible at all. Observations of events with amplitudes in between those of large dSpikes and small

dSpikes (spikelets) are rare, because the transition from a large dSpike to a small dSpike occurs over very short distances due to the large attenuation at branch points (*Katz et al., 2009*). Accordingly, we observed a large gap in the distribution of event amplitudes, with no events having amplitudes in the 30–40 mV range (*Figure 2—figure supplement 1A*; *Figure 2—source data 2*). Small dSpikes (spikelets) were distinguishable from the cases without any regenerative events (i.e., EPSPs only) because they had larger dV/dt values (*Figure 2B* and *Figure 2—figure supplement 1B*; *Figure 2—source data 2*; see below and 'Materials and methods').

In the presence of low TTX, both the number and the amplitude of large-amplitude dendritic events (i.e., bAPs and large dSpikes, all with amplitudes >40 mV; see 'Materials and methods') were reduced (control: amplitude = 56.2 ± 1.8 mV, n = 39 events; low TTX: amplitude = 44.8 ± 1.7 mV, n = 17 events; p < $10^{-4}$ by Student's t-test). Although distinguishing between bAPs and large dSpikes was only possible in some cases (*Figure 2—figure supplement 2*; *Figure 2—source data 2*; see 'Materials and methods'), the two events that were the most clearly large dSpikes were reduced to spikelets in response to the same stimulus in the presence of low TTX (*Figure 2A*).

Of the smaller events (i.e., <30 mV), some small dSpikes were relatively easy to identify in dendritic recordings (e.g., *Figure 2B*), because they were the most extreme outliers in the distribution of event dV/dt values. Other small dSpikes were more difficult to resolve because their attenuation from the site of origin makes them barely distinguishable from the cases without regenerative events (i.e., EPSPs only). Thus, rather than using a single definition for small dSpikes, we employed a range of definitions; then for each definition we compared the control and low TTX conditions. In this way, we could determine whether conclusions about the effects of low TTX on dSpikes were dependent on the definition used.

We varied the dV/dt criterion for small dSpikes from a relatively inclusive one (outside the 85% prediction band of a fit to the distribution of all events normalized dV/dt in control and low TTX; *Figure 2C*; see 'Materials and methods') to a more restrictive one (outside the 99% prediction band). We found that application of low TTX dramatically reduced the number of small dSpikes defined by all confidence levels in this range (*Figure 2C,D*; *Figure 2—source data 1*; see *Figure 2—figure supplement 3* for all small dSpikes identified by the 99% confidence level; *Figure 2—source data 2*). Similar results were obtained using other methods for identifying small dSpikes (*Figure 2—figure supplement 4*; *Figure 2—source data 2*; see 'Materials and methods'). As both large dSpikes and small dSpikes (spikelets) reflect the presence of dSpikes initiated in the tuft dendrites, and both were inhibited by low TTX, we conclude that low TTX is an effective inhibitor of dSpikes generated in response to TBS of the PP.

Given the ability of low TTX to inhibit dendritically initiated spikes, we tested its effects on the induction of LTP using TBS at PP → CA1$_{tuft}$ synapses (using stimulus intensities yielding single EPSPs of 2–5 mV in somatic recordings). TBS was repeated three times (TBSx3) and was either paired with brief somatic current injections (TBSx3+Current) to evoke action potential firing during each burst of presynaptic stimuli (*Figure 3A*; *Figure 3—source data 1*) or delivered with somatic voltage clamp (TBSx3+SomaticVC) to prevent somatic action potential firing (holding potential −68 to −70 mV; see 'Materials and methods'). The magnitude of PP → CA1$_{tuft}$ LTP was similar in these two conditions (*Figure 3B–E*; TBSx3+Current, potentiation ratio = 1.63 ± 0.16, n = 9; TBSx3+SomaticVC, potentiation ratio = 1.53 ± 0.07, n = 9; p = 1.00 by one-way ANOVA; *Figure 3—source data 1*), suggesting that action potential firing did not contribute to LTP induction at these synapses (see also *Golding et al., 2002*). Under both conditions, PP → CA1$_{tuft}$ LTP was blocked completely in the presence of low TTX (*Figure 3C–E*; *Figure 3—source data 1*).

Because low TTX blocks both dSpikes and bAPs, it is conceivable that TTX inhibits LTP by blocking either or both of these sources of postsynaptic depolarization. However, consistent with our previous study (*Golding et al., 2002*), we showed here again that PP → CA1$_{tuft}$ LTP is not affected by blocking bAPs. Thus, we attribute the effect of low TTX on this particular form of LTP to its effect on Na-dSpikes. This is a surprising result, because of the brief duration of Na-dSpikes, compared to Ca- or NMDA-dSpikes.

One possible explanation for the requirement of Na-dSpikes during PP → CA1$_{tuft}$ LTP induction is that they are required to activate channels mediating synaptic calcium influx. We therefore sought to determine which calcium-permeable channels are responsible for the underlying sources of calcium influx. We previously showed that PP → CA1$_{tuft}$ LTP was reduced ~50% by blocking NMDARs, ~50% by blocking L-, T-, and R-type Ca$_v$ channels, and almost completely by blocking NMDARs and Ca$_v$ channels together (*Golding et al., 2002*). We confirmed these results here and showed further that the effects of Ca$_v$ channel blockers were attributable to blocking L-type Ca$_v$ channels (with 10 μM nimodipine), but not blocking T-, R-, or other types of Ca$_v$ channels (with 50 μM Ni$^{2+}$; *Figure 4*; *Figure 4—source data 1*;

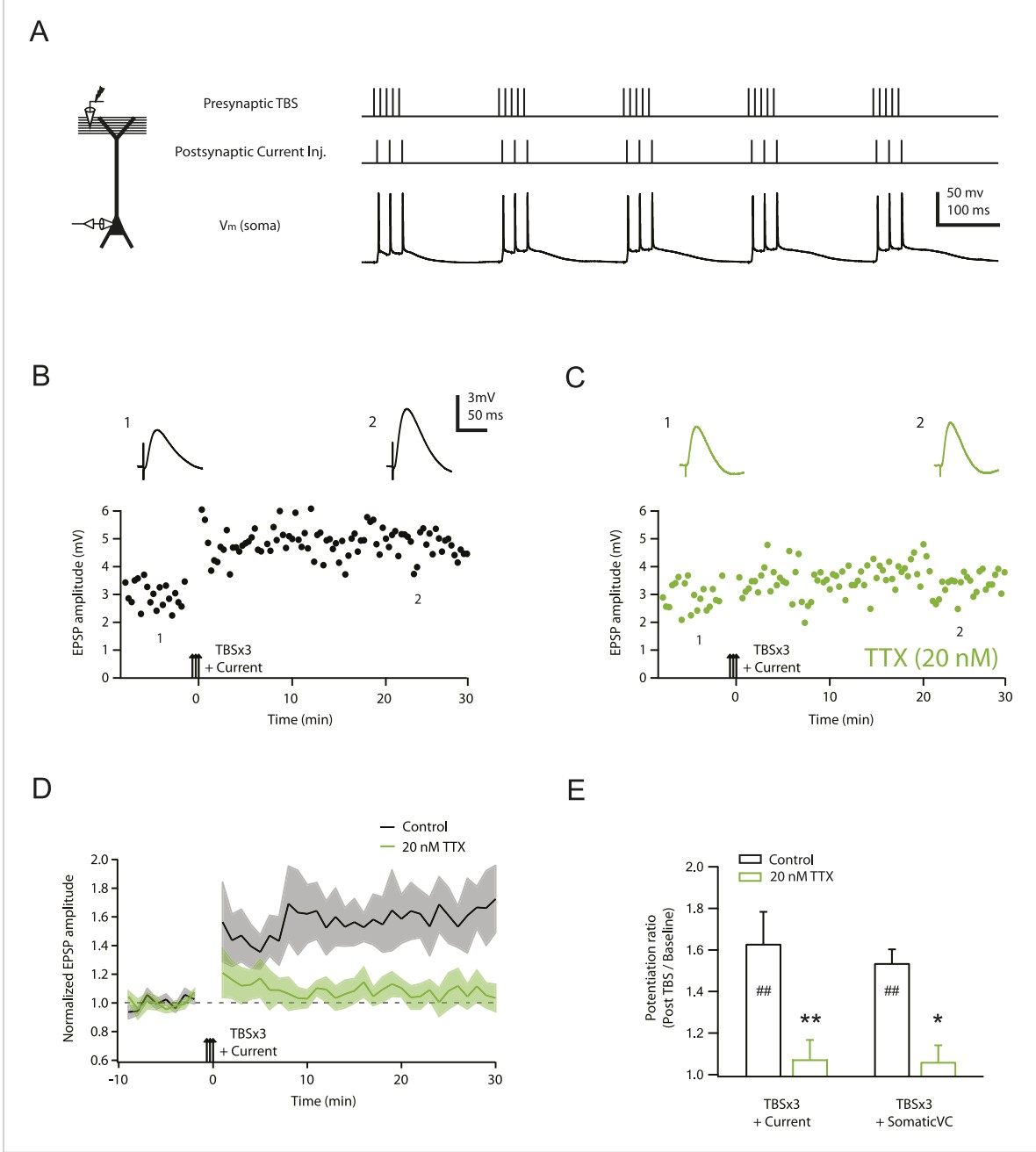

**Figure 3**. TBS-induced PP → CA1$_{tuft}$ LTP is blocked by reducing Na$_v$ channel availability with 20 nM TTX. (**A**) Left, experimental configuration showing somatic whole-cell recording with presynaptic stimulation of the PP. Right, representative trace of somatically recorded voltage in response to presynaptic TBS paired with somatic current injections at 50 Hz. A TBS consisted of five high-frequency bursts repeated at 5 Hz, with each consisting of 5 stimuli at 100 Hz. Each TBS was delivered three times, at 30-s intervals. (**B**, **C**) Representative time course of EPSP amplitude before and after TBS was delivered three times, paired with somatic current injections (TBSx3+Current; arrows) in control (**B**) and 20 nM TTX (**C**; 20 nM TTX was applied via the bath during the entire experiment). Top, representative traces (single trials) of EPSP before (1) and 25 min after (2) TBSx3+Current were delivered. The scale bar in **B** applies to all panels. (**D**) Summary of the LTP experiments in control and 20 nM TTX. EPSP amplitude is normalized to the average EPSP amplitude before LTP induction. Solid lines and shaded areas represent mean and S.E.M., respectively. (**E**) Potentiation ratio in different experimental conditions: TBSx3+ Current, TBS paired with somatic current injections; TBSx3+SomaticVC, TBS delivered with the soma voltage-clamped (VC) at ∼−70 mV to prevent action potential firing (TBSx3+Current, n = 9; TBSx3+SomaticVC, n = 9; TBSx3+Current in 20 nM TTX, n = 8; TBSx3+SomaticVC in 20 nM TTX, n = 8). ##p < 0.01 for the effect of time on EPSP amplitude by one-way repeated measures ANOVA. **p < 0.01 (TBSx3+Current, control vs 20 nM TTX), *p < 0.05 (TBSx3+ SomaticVC, control vs 20 nM TTX) by one-way ANOVA with post-hoc means comparison using Tukey's test.

The following source data is available for figure 3:

**Source data 1**. Source data of *Figure 3*.

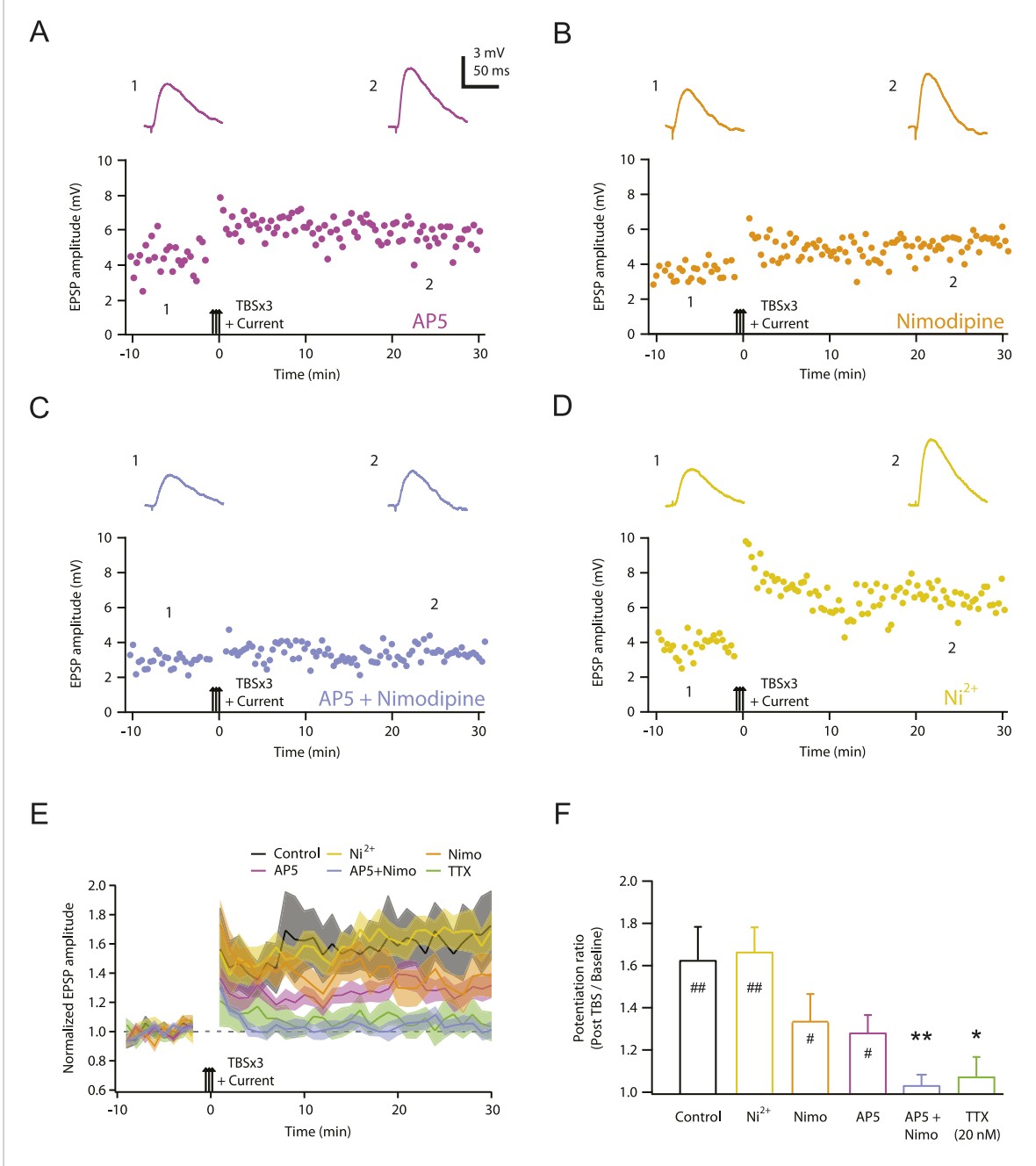

**Figure 4**. NMDAR and L-Ca$_v$ channels mediate TBS-induced LTP at PP → CA1$_{tuft}$ synapses. (**A–D**) Representative time course of EPSP amplitude before and after TBSx3+Current was delivered (arrows) in 50 µM AP5 (**A**), 10 µM nimodipine (**B**), 50 µM AP5 and 10 µM nimodipine (**C**), or 50 µM Ni$^{2+}$ (**D**). Drugs were applied via the bath during the entire experiment. Top, representative traces (single trials) of EPSP before (1) and 25 min after (2) TBSx3+Current was delivered. The scale bar in **A** applies to all panels. (**E**) Summary of the LTP experiments in AP5, nimodipine (Nimo), AP5 and nimodipine (AP5+Nimo), and Ni$^{2+}$, shown along with the data in control and 20 nM TTX (*Figure 3*) for comparison. EPSP amplitude is normalized to the average EPSP amplitude before LTP induction. Solid lines and shaded areas represent mean and S.E.M., respectively. (**F**) Potentiation ratio in different experimental conditions (Control, n = 9; AP5, n = 11; Nimo, n = 9; AP5+Nimo, n = 11; Ni$^{2+}$, n = 9; 20 nM TTX, n = 8). ##p < 0.01, #p < 0.05 for the effect of time on EPSP amplitude by one-way repeated measures ANOVA. **p < 0.01, *p < 0.05 (all compared to control) by one-way ANOVA with post-hoc means comparison using Tukey's test.

The following source data and figure supplement are available for figure 4:

**Source data 1**. Source data of *Figure 4*.

*Figure 4. continued on next page*

*Figure 4. Continued*

**Source data 2**. Source data of *Figure 4—figure supplement 1*.

**Figure supplement 1**. L-Ca$_v$ channels are not required for synaptic transmission before or after the induction of PP → CA1$_{tuft}$ LTP.

see also Remy and Spruston, 2007). Furthermore, 10 µM nimodipine had no effect on synaptic responses, either before or after the induction of PP → CA1$_{tuft}$ LTP (*Figure 4—figure supplement 1*; *Figure 4—source data 2*), suggesting that L-type Ca$_v$ channels are involved in the induction rather than the expression of LTP.

These results suggest that NMDAR and L-type Ca$_v$ (L-Ca$_v$) channels are the two main sources of calcium entry during LTP induction by TBS of PP → CA1$_{tuft}$ synapses. The observation that partial block of Na$_v$ channels by low TTX blocked PP → CA1$_{tuft}$ LTP almost completely (*Figure 3B–E*) indicates that Na$_v$ channel-mediated events are required to activate these two types of calcium-permeable channels, and these events are unlikely to be bAPs, as we showed here and in our previous study (see above). We therefore performed additional analyses on the dendritic recordings (described above) to determine how low TTX affects membrane potential changes that can activate these channels.

As described above, low TTX blocked large and small dSpikes (*Figure 2* and *Figure 2—figure supplement 4*). In one of the two dendritic recordings where we observed inhibition of a clear large dSpike by low TTX, we subsequently washed out TTX and applied 50 µM AP5, which did not block the large dSpike (*Figure 5A*; *Figure 5—source data 1*). By contrast, low TTX had almost no effect on the slow depolarization produced by each burst of PP → CA1$_{tuft}$ EPSPs in response to TBS, whereas 50 µM AP5 had a much greater effect on the slow depolarization associated with each burst. This differential effect of low TTX and 50 µM AP5 on the slow depolarization was confirmed in additional dendritic recordings (n = 9; for four cases, data in TTX and AP5 were obtained from the same cell; *Figure 5B*; *Figure 5—source data 1*). Because low TTX was a more potent inhibitor of PP → CA1$_{tuft}$ LTP than 50 µM AP5 (*Figure 4F*), these results suggest that the fast depolarization associated with Na-dSpikes is more important for the induction of PP → CA1$_{tuft}$ LTP than the slow synaptic depolarization mediated solely by EPSPs.

To test this conclusion further, we designed a stimulation protocol expected to have differential effects on Na-dSpikes and slow synaptic depolarization. A modified TBS pattern consisting of only 2 stimuli in each burst (2-stim TBSx3, *Figure 6A*; *Figure 6—source data 1*) is expected to greatly reduce the slow synaptic depolarization while preserving about 50% of Na-dSpikes (based on *Figure 2C*). This 2-stim protocol induced LTP at PP → CA1$_{tuft}$ synapses of approximately the same magnitude as various TBSx3 protocols consisting of 5 stimuli (5-stim) in each burst (*Figure 6B,C*; *Figure 6—source data 1*). The 2-stim TBSx3 protocol yielded PP → CA1$_{tuft}$ LTP of 82% of the average LTP induced by all variants of the 5-stim TBSx3 protocol, despite a greatly reduced integral of the slow depolarization (27% of that with 5-stim TBSx3; *Figure 6D,E*; *Figure 6—source data 1*).

Collectively, the results above indicate that low TTX blocks PP → CA1$_{tuft}$ LTP by inhibiting fast Na-dSpikes, with minimal effect on the slow synaptic depolarization induced by each high-frequency burst during TBS. We next turned our attention to investigating the calcium entry that links Na-dSpikes with the induction of PP → CA1$_{tuft}$ LTP. Given that NMDAR and L-Ca$_v$ channels are both voltage-dependent sources of calcium entry, and they both contribute to PP → CA1$_{tuft}$ LTP (approximately equally, see *Figure 4F*), the most parsimonious hypothesis is that Na-dSpikes are effective activators of calcium entry via these pathways. Although LTP induction at these synapses may depend on the complex spatiotemporal properties of calcium entry, the simplest scenario is that Na-dSpikes produce particularly large calcium entry in synaptically activated portions of the tuft dendrites, which triggers LTP.

To test this idea, CA1 pyramidal neurons were filled with 100 µM Oregon Green 488 BAPTA-1 (OGB-1) and 50 µM Alexa Fluor 594 Hydrazide (AF-594) and imaged on a two-photon laser-scanning microscope during stimulation of the PP with single burst of 5 stimuli at 100 Hz ('Materials and methods'). In these experiments, the soma was voltage-clamped at ∼ −70 mV in order to prevent action potential firing, thus limiting detected calcium influx to that induced by synaptic events and dendritically initiated spikes. For quantification of calcium signals, the calcium-induced change in OGB-1 fluorescence was measured as a fraction of the baseline fluorescence of the calcium-insensitive reference, AF-594 (ΔG/R; see 'Materials and methods'). Although we could easily resolve dendritic

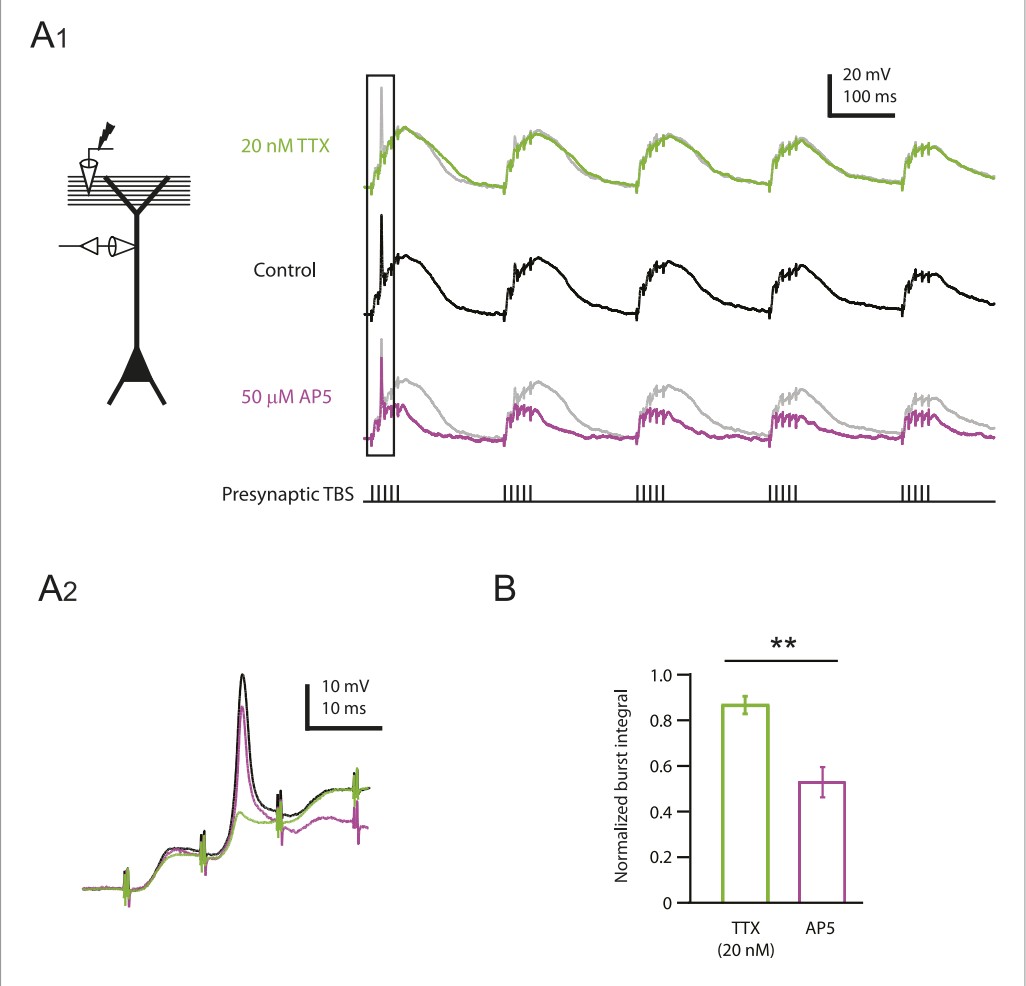

**Figure 5**. Reducing Na_v channel availability reduces the slow synaptic depolarization in distal apical trunk in response to TBS significantly less than blocking NMDAR channels. (**A**) Differential effects of 20 nM TTX and 50 µM AP5 on dendritically recorded voltage in response to presynaptic TBS. **A$_1$**, Left, experimental configuration showing dendritic whole-cell recording with presynaptic stimulation of the PP. Right, example traces of dendritically recorded voltage in response to presynaptic TBS from a neuron with a large dSpike in control (the same cell as the *bottom* case shown in *Figure 2A*) and the corresponding responses in 20 nM TTX or 50 µM AP5 (TTX was washed out before subsequent application of AP5). **A$_2$**. Traces corresponding to the box in **A$_1$**. Note that the large dSpike was blocked by 20 nM TTX, but not 50 µM AP5. (**B**) Summary of effects of TTX (n = 9) and AP5 (n = 4) on burst responses (normalized to control). Measured integrals include contributions from large-amplitude dendritic events, which were observed in some cells. **p < 0.01 by Student's t-test.

The following source data is available for figure 5:

**Source data 1**. Source data of *Figure 5*.

---

spines, we could not determine which spines received direct synaptic input, so we restricted our analysis to the calcium signals in dendritic shafts (*Figure 7A*), using the response integral as the best measure of dendritic calcium entry (see 'Materials and methods'; *Figure 7—figure supplement 1*; *Figure 7—source data 2*).

Burst stimulation of the PP resulted in clear ΔG/R responses (79 ± 14%, n = 17 cells), which were significantly reduced by 50 µM AP5 (*Figure 7B*; *Figure 7—source data 1*), indicating that we were able to detect calcium entry dependent on synaptic activation—likely mediated by multiple channel types—when imaging from dendritic shafts. Importantly, however, low TTX was far less potent at inhibiting intracellular calcium elevations than 50 µM AP5 (*Figure 7B,C*; *Figure 7—source data 1*).

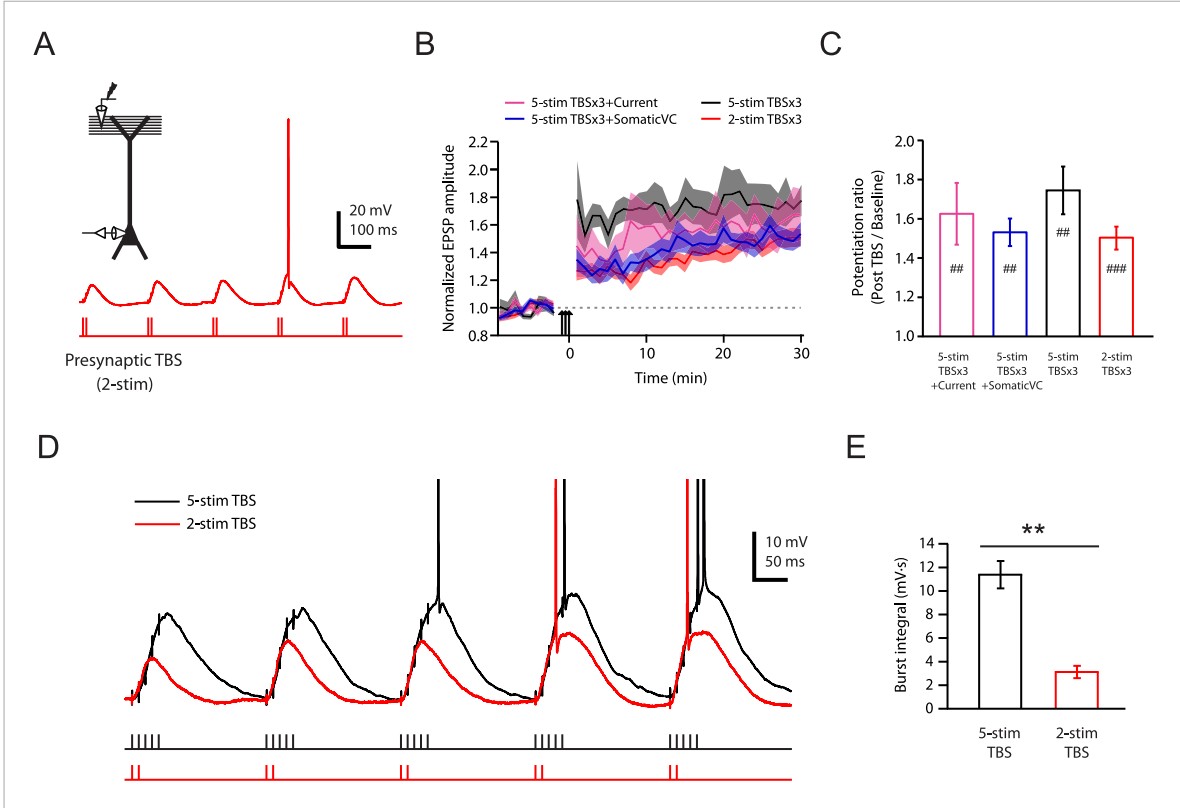

**Figure 6**. Brief synaptic stimuli are sufficient for the induction of PP → CA1$_{tuft}$ LTP. (**A**) Top left, experimental configuration showing somatic whole-cell recording with presynaptic stimulation of the PP. Bottom right, representative trace of somatically recorded voltage in response to a modified TBS pattern, consisting of only 2 (2-stim TBS) instead of 5 (5-stim TBS) synaptic stimuli in each burst. (**B**) Summary of the time courses of normalized EPSP amplitude before and after 2-stim TBS or 5-stim TBS was delivered three times (2-stim TBSx3 or 5-stim TBSx3; arrows), shown along with the data with TBSx3+Current or TBSx3+SomaticVC (**Figure 3**) for comparison. EPSP amplitude is normalized to the average EPSP amplitude before LTP induction. Solid lines and shaded areas represent mean and S.E.M., respectively. (**C**) Potentiation ratio in different experimental conditions (5-stim TBSx3+Current, n = 9; 5-stim TBSx3+SomaticVC, n = 9; 5-stim TBSx3, n = 5; 2-stim TBSx3, n = 13). ###p < 0.001, ##p < 0.01 for the effect of time on EPSP amplitude by one-way repeated measures ANOVA. No statistically significant differences by one-way ANOVA (p = 0.45). (**D**) Representative traces of somatically recorded voltage in response to 5-stim or 2-stim TBS. The action potentials are truncated. (**E**) Summary of the integral of responses to the first repeat of 5-stim (n = 5) or 2-stim (n = 13) TBS. **p < 0.01 by Student's t-test.

The following source data is available for figure 6:

**Source data 1**. Source data of **Figure 6**.

This result was surprising, given that it is opposite to what one would expect from the result that low TTX was a far more potent inhibitor of PP → CA1$_{tuft}$ LTP than 50 µM AP5. This suggests that the effectiveness of low TTX as a blocker of LTP cannot be explained by its effect on the measured dendritic calcium signals.

To assist with the interpretation of our results, we used a compartmental model of a CA1 pyramidal neuron. The model was adapted from a previous model that contained Na$_v$ and K$_v$ channels in the dendrites to reproduce experimental data on bAPs and dSpikes (**Golding et al., 2001**; **Jarsky et al., 2005**; **Katz et al., 2009**); it was extended to include NMDARs at all synapses and L-Ca$_v$ channels (high-voltage-activated Ca$_v$ channels) in the dendrites, as well as simple models of calcium buffers (including endogenous buffer and OGB-1), calcium diffusion, and calcium extrusion ('Materials and methods').

The model was able to reproduce the effects of low TTX (simulated by 50% reduction of Na$_v$ channel conductance) and 50 µM AP5 (simulated by complete block of NMDARs) on dSpikes and EPSPs recorded from the distal apical trunk (**Figure 8A–C**; **Figure 8—source data 1**). The simulations were also able to reproduce the effects of TTX and AP5 on dendritic calcium transients (**Figure 7**), as

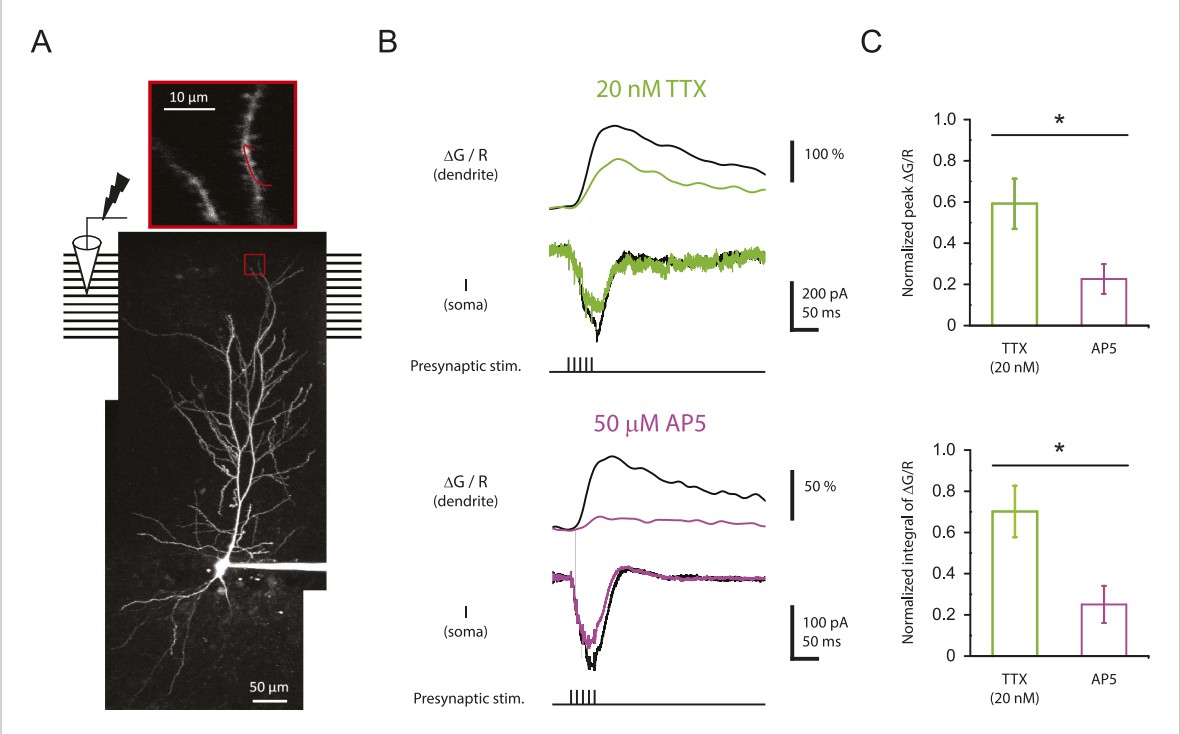

**Figure 7**. Reducing Na$_v$ channel availability reduces the calcium influx in distal apical tuft dendrites in response to high-frequency burst stimulation significantly less than blocking NMDAR channels. (**A**) Bottom, experimental configuration showing somatic whole-cell recording with presynaptic stimulation of the PP, with a representative Z-stack image of a neuron (filled with 50 μM AF-594) on which the imaging experiments were performed. The box indicates the field of view used during high-resolution line scan. Top, a single-scan image corresponding to the box in the image below. The red line represents a line profile used for line scan, which went through a short stretch of the dendritic shaft and crossed a neighboring spine (not shown). Analysis was restricted to the data from dendritic shafts only. (**B**) Representative traces of ΔG/R (100 μM OGB-1) from dendritic shaft and current ('I') from somatic voltage-clamp recording in response to a single burst stimulation in control and 20 nM TTX or 50 μM AP5 (at least 15 min after drug application). Note that a spikelet was blocked by 20 nM TTX. (**C**) Summary of effects of TTX (n = 7) and AP5 (n = 5) on ΔG/R (normalized to control). *p < 0.05 by Student's t-test.

The following source data and figure supplement are available for figure 7:

**Source data 1**. Source data of *Figure 7*.

**Source data 2**. Source data of *Figure 7—figure supplement 1*.

**Figure supplement 1**. Stability of two-photon calcium imaging over long recording durations.

measured by the modeled increase in the concentration of simulated calcium-bound OGB-1 ([Ca$^{2+}$]$_{OGB}$) in response to burst activation of simulated synapses (*Figure 8B,C*; *Figure 8—source data 1*). The ability of the model to reproduce these experimental observations allowed us to use it to explore the possible mechanism by which Na-dSpikes mediate calcium influx that may be critical for the induction of PP → CA1$_{tuft}$ LTP.

In response to burst activation of simulated synapses a single large dSpike was observed in the apical trunk. However, in small apical tuft branches, where direct patch-clamp recording cannot be performed, simulations revealed that a dSpike was initiated on several branches in response to burst activation of spatially distributed synaptic inputs, appearing as multiple dSpikes on the apical tuft branch (*Figure 8D*; *Figure 8—source data 1*). This is consistent with the observations in the experiments that multiple spikelets occurred in response to burst stimulation (see *Figure 2—figure supplement 3*).

The simulations also offered insight into how Na-dSpikes could provide the calcium influx necessary for PP → CA1$_{tuft}$ LTP induction. By comparing the dendritic calcium signals (reported by simulated OGB-1) and the calcium influx through the synaptic channels in the model, we could clearly distinguish the difference between dendritic 'bulk' calcium signals and the calcium signals near the

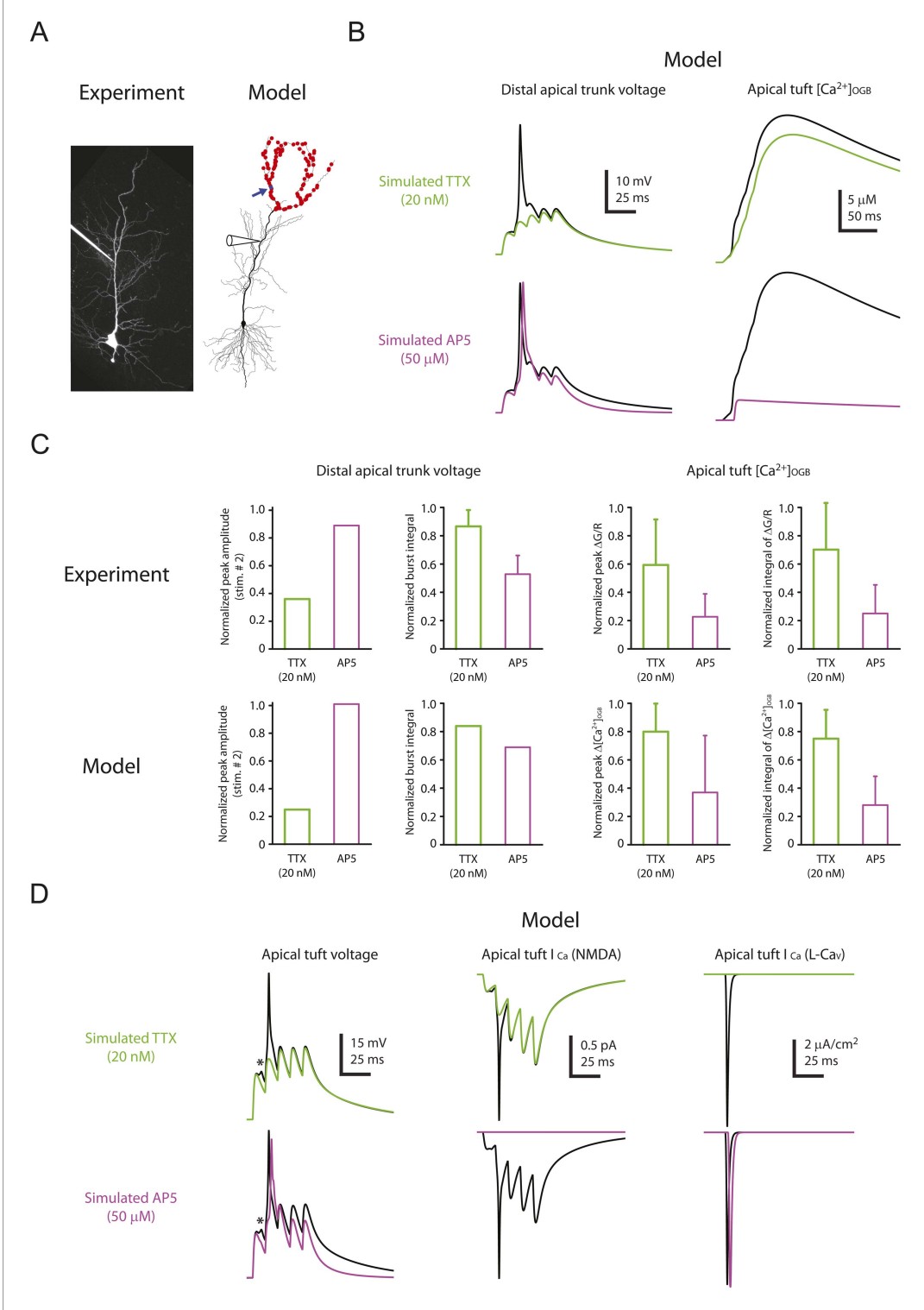

**Figure 8**. Computational modeling reveals a large, rapid, Na-dSpike-mediated calcium current through NMDAR and L-Ca$_v$ channels. (**A**) Left, Z-stack image of a neuron (filled with 50 μM AF-594). Right, morphology of the model neuron. Red dots indicate the locations of simulated synapses. Arrow indicates the imaged compartment with a simulated synapse (blue dot) for the example shown in **B** and **D**. (**B**) Representative traces of simulated voltage recorded from the distal apical trunk and concentration of calcium-bound OGB-1 ([Ca$^{2+}$]$_{OGB}$) recorded from a compartment with a simulated synapse on the distal apical tuft (as indicated in **A**) in response to a single burst

*Figure 8. continued on next page*

*Figure 8. Continued*

stimulation in control and simulated 20 nM TTX or 50 µM AP5. (**C**) Summary of the simulations and a comparison with the experiments in *Figures 5, 7*. The bar graphs show mean ± S.D. (normalized to control; S.D. was used to emphasize the distribution and variability of the drug effects across different imaging locations). For normalized peak amplitude, only the cell with a dSpike initiated presumably the closest to and actively propagating to the recording electrode in our dendritic recordings (*Figure 5A*) is illustrated for comparison. For normalized peak $\Delta[Ca^{2+}]_{OGB}$ and normalized integral of $\Delta[Ca^{2+}]_{OGB}$, data were pooled from all the compartments on the distal apical tuft (the number of compartments = 440), to mimic the experimental condition in which the imaging locations were arbitrarily selected. (**D**) Representative traces of simulated voltage and instantaneous fractional calcium current through NMDAR channels ($I_{Ca}$, NMDA) and L-$Ca_v$ channels ($I_{Ca}$, L-$Ca_v$). Modeled voltages and currents are from a compartment with a simulated synapse on the distal apical tuft (as indicated in **A**) in response to a single burst stimulation in control and simulated 20 nM TTX or 50 µM AP5. Note the presence of a spikelet (asterisk), which was due to a Na-dSpike initiated in a neighboring dendritic branch and failing to propagate reliably to the recorded branch.

The following source data and figure supplement are available for figure 8:

**Source data 1**. Source data of *Figure 8*.
**Source data 2**. Source data of *Figure 8—figure supplement 1*.
**Figure supplement 1**. Minimal effect of blocking L-$Ca_v$ channels on the calcium influx in distal apical tuft dendrites is consistent with the prediction of the model.

mouth of channels. Simulations at the synapses revealed that the largest instantaneous calcium currents ($I_{Ca}$) through NMDAR and L-$Ca_v$ channels were mediated by locally generated Na-dSpikes, owing to efficient relief of magnesium block of NMDAR channels and strong activation of L-$Ca_v$ channels, respectively (*Figure 8D*; *Figure 8—source data 1*). These simulations, combined with the experimental results, suggest that a large, rapid, Na-dSpike-mediated component of calcium entry may produce transient, localized increases in intracellular calcium concentration that are essential for the induction of TBS-induced PP → $CA1_{tuft}$ LTP. To further test this model, we experimentally tested two predictions.

First, we posited that blocking L-$Ca_v$ channels, which inhibits LTP by ~50%, would have minimal effect on measurements of bulk calcium entry in the distal apical dendrites, as indicated in our model (*Figure 8—figure supplement 1D–F*; *Figure 8—source data 2*; see also *Tsay et al., 2007*). Indeed, we found that 10 µM nimodipine did not have a significant effect on dendritic calcium responses during burst stimulation of the PP (*Figure 8—figure supplement 1A–C*; *Figure 8—source data 2*). This result is consistent with our hypothesis that L-$Ca_v$ channels contribute to the induction of PP → $CA1_{tuft}$ LTP by mediating increases in localized calcium near the channel pores rather than in dendritic bulk calcium.

Second, we posited that LTP would be inhibited by chelating intracellular calcium, but only when the buffering kinetics were fast enough to disrupt localized calcium signaling close to the channel pores (see 'Discussion'). 1,2-*Bis*(2-aminophenoxy)ethane-*N*,*N*,*N'*,*N'*-tetraacetic acid (BAPTA) has a much faster calcium binding kinetics than ethylene glycol-*bis*(2-aminoethylether)-*N*,*N*,*N'*,*N'*-tetraacetic acid (EGTA) (*Tsien, 1980*), and we therefore tested this hypothesis by buffering cytoplasmic calcium elevations with either a relatively low or high concentration of EGTA or BAPTA (*Maravall et al., 2000*; *Henneberger et al., 2010*), while maintaining the basal calcium concentration at 50 nM by adding an appropriate amount of $CaCl_2$ ('Materials and methods'). We predicted that only a high concentration of BAPTA should buffer intracellular calcium increases rapidly and effectively enough to block PP → $CA1_{tuft}$ LTP, and this was indeed found to be the case (*Figure 9*; *Figure 9—source data 1*). These results suggest that NMDAR and L-$Ca_v$ channels contribute to the induction of TBS-induced PP → $CA1_{tuft}$ LTP by mediating calcium influx, and are consistent with the hypothesis that it is the large, transient, localized increases in intracellular calcium concentration mediated by Na-dSpikes that are critical for this form of LTP.

## Discussion

Our strategy of partially blocking $Na_v$ channels with a low concentration of TTX was designed to test the hypothesis that Na-dSpikes are important contributors to the postsynaptic depolarization required for TBS-induced LTP of PP synapses in the distal tuft of CA1 pyramidal neurons. The results

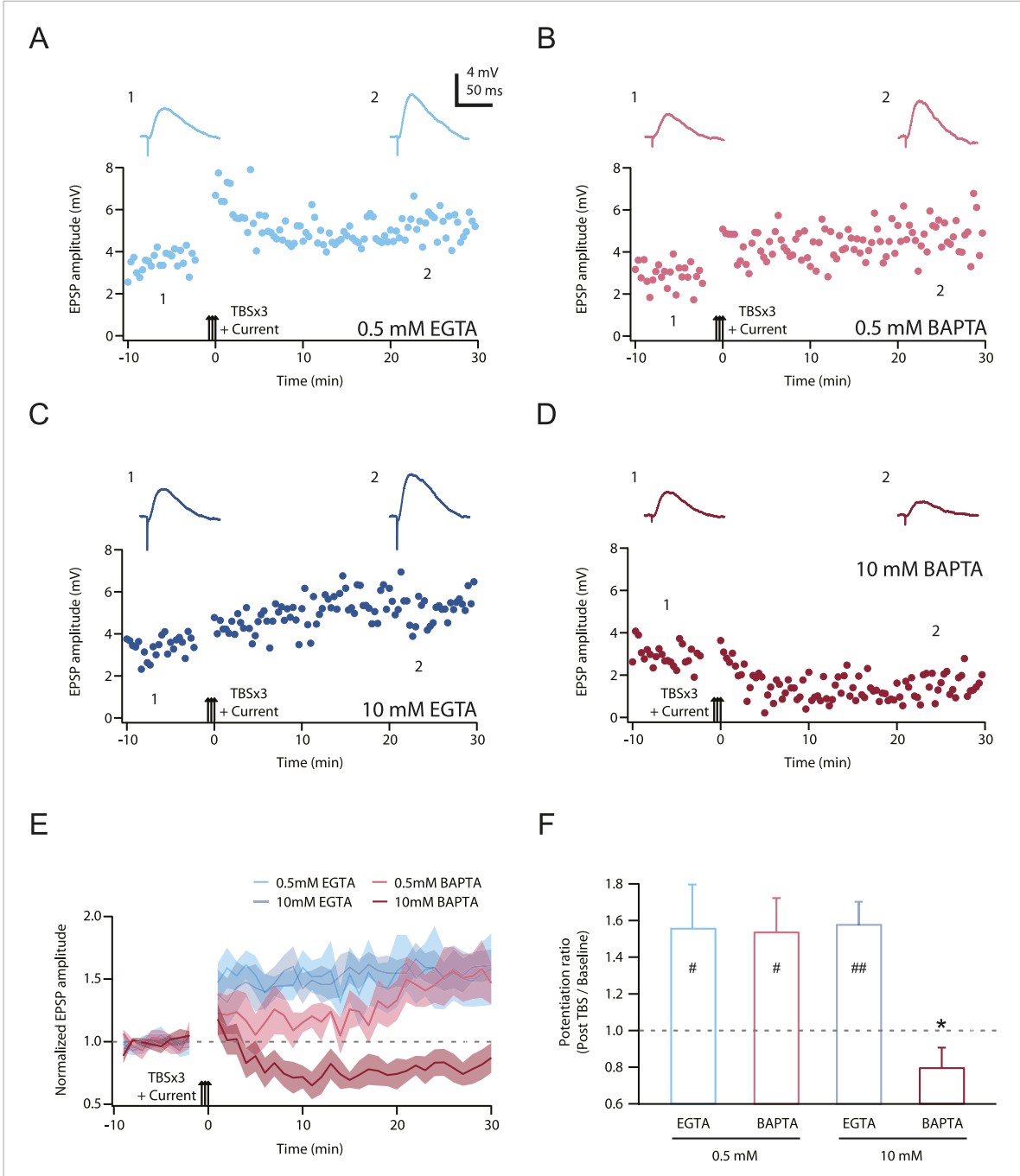

**Figure 9**. TBS-induced PP → CA1$_{tuft}$ LTP is blocked by intracellular calcium buffering with a high concentration of BAPTA, but not a low concentration of BAPTA or low or high EGTA. (**A–D**) Representative time course of EPSP amplitude before and after TBSx3+Current was delivered (arrows) from cells buffered with 0.5 mM EGTA (**A**), 0.5 mM BAPTA (**B**), 10 mM EGTA (**C**), or 10 mM BAPTA (**D**). CaCl$_2$ was added to maintain basal calcium level (∼50 nM; see 'Materials and methods'). Calcium buffer was included in the intracellular solution. Top, representative traces (single trials) of EPSP before (1) and 25 min after (2) TBSx3+Current was delivered. The scale bar in **A** applies to all panels. (**E**) Summary of the LTP experiments with different calcium buffering. EPSP amplitude is normalized to the average EPSP amplitude before LTP induction. Solid lines and shaded areas represent mean and S.E.M., respectively. (**F**) Potentiation ratio in different experimental conditions (0.5 mM EGTA, n = 8; 0.5 mM BAPTA, n = 5; 10 mM EGTA, n = 5; 10 mM BAPTA, n = 8). ##$p < 0.01$, #$p < 0.05$ for the effect of time on EPSP amplitude by one-way repeated measures ANOVA. *$p < 0.05$ (compared to 10 mM EGTA) by one-way ANOVA with *post hoc* means comparison using Tukey's test.

The following source data is available for figure 9:

**Source data 1**. Source data of *Figure 9*.

support this hypothesis, as this manipulation resulted in block of PP → CA1$_{tuft}$ LTP. The completeness of the block further suggests that Na-dSpikes are a critical component of the LTP induction process at these synapses.

## Key findings and interpretation

Our results advance our understanding of the mechanisms underlying PP → CA1$_{tuft}$ LTP induction in three important ways. First, whereas previous evidence for the role of dSpikes in the induction of PP → CA1$_{tuft}$ LTP (*Golding et al., 2002*) was correlative (stimuli that evoked dSpikes also induced LTP, while stimuli that failed to evoke dSpikes also failed to induce LTP), this work supports a more direct causal link between the occurrence of dSpikes and the induction of PP → CA1$_{tuft}$ LTP.

Second, our results firmly establish a role for dendritic Na$_v$ channels in the induction of PP → CA1$_{tuft}$ LTP. Previously observed correlations between dSpikes and PP → CA1$_{tuft}$ LTP (*Golding et al., 2002*) could not discern whether the relevant dSpikes were mediated by Na$_v$, Ca$_v$, or NMDAR channels. One possibility is that these channels could be activated in concert and support relevant dendritic plateau potentials via a 'spike-chain mechanism' (*Schiller et al., 2000*) to trigger PP → CA1$_{tuft}$ LTP, but our results suggest that Na-dSpikes are the major contributors, and slow NMDAR-dependent synaptic depolarization is neither sufficient nor necessary for LTP at these synapses (*Figures 5, 6*). Our evidence for the necessity of Na-dSpikes in synaptic plasticity adds to the list of important functions of dendritic Na$_v$ channels, which also includes contributions to synaptic integration leading to action potential initiation and a central role in action potential backpropagation (*Stuart and Sakmann, 1994*; *Golding et al., 2001*; *Larkum and Nevian, 2008*; *Spruston, 2008*).

Third, our results provide insight into the mechanisms by which Na-dSpikes are coupled to the induction of LTP through calcium influx at PP → CA1$_{tuft}$ synapses, as discussed in detail below. The block of LTP only by the fast calcium chelator BAPTA strongly supports the hypothesis that the large, localized increases in intracellular calcium concentration are necessary for the induction of PP → CA1$_{tuft}$ LTP. This is in interesting contrast to the block of LTP also by the slow calcium chelator EGTA at the more proximal Schaffer collateral synapses of CA1 pyramidal neurons (*Lynch et al., 1983*), which suggests a different underlying mechanism for the calcium dependence of LTP at these synapses.

The most parsimonious explanation for our results is that the effective block of LTP by low TTX is through the inhibition of Na-dSpikes per se; however, a few alternative interpretations should be considered. For example, LTP induction could depend on a non-ionic mechanism directly coupled to the conformational changes of Na$_v$ channels or an ionic mechanism mediated by sodium flux through them (but little has been reported regarding these speculative mechanisms). Alternatively, during synaptic activation, Na$_v$ channels may affect membrane potential and calcium entry in spines independently of dSpikes (*Araya et al., 2007*; *Bloodgood and Sabatini, 2007*; note that blocking Na$_v$ channels enhanced synaptic calcium influx in *Bloodgood and Sabatini, 2007*). Finally, TTX may have a presynaptic effect, as presynaptic sodium has been shown to regulate synaptic strength constitutively at the Calyx of Held (*Huang and Trussell, 2014*). A few observations argue against these possibilities, such as the lack of effect of low TTX on synaptic responses at PP → CA1$_{tuft}$ synapses (*Figure 1*), the strong correlation between dSpikes and PP → CA1$_{tuft}$ LTP (*Golding et al., 2002*) and the reduction in Na-dSpikes that accompanies the block of PP → CA1$_{tuft}$ LTP by low TTX (*Figures 2, 3*). However, additional experiments would be required to test these alternative possibilities directly.

## Mechanisms for coupling Na-dSpikes to the induction of PP → CA1$_{tuft}$ LTP

The finding that Na-dSpikes play such a critical role in the induction of PP → CA1$_{tuft}$ LTP is somewhat surprising, given the brief duration of these Na-dSpikes compared to slow depolarization mediated by EPSPs, Ca-dSpikes, or NMDA-dSpikes. The fact that simultaneously blocking NMDAR and L-Ca$_v$ channels eliminated PP → CA1$_{tuft}$ LTP (*Golding et al., 2002*; *Remondes and Schuman, 2003*; *Ahmed and Siegelbaum, 2009*) suggests that Na-dSpikes contribute to the induction of PP → CA1$_{tuft}$ LTP by increasing calcium influx through these channels. But how does this work, given that Na-dSpikes have a brief duration and thus a limited contribution to the increase in 'bulk' calcium concentration (i.e., the calcium measurable with the spatiotemporal resolution provided by imaging with calcium-sensitive dyes) compared to slow synaptic depolarization?

A simple explanation, supported by our modeling (**Figure 8**) and the subsequent experiments (**Figure 9**), is that the calcium influx during Na-dSpikes (via NMDAR and L-Ca$_v$ channels) is vigorous and brief, whereas the calcium influx during slow synaptic depolarization (via NMDAR channels) is smaller, but longer lasting. Although the latter ultimately contributes more to the total amount of calcium entry during a burst and comprises the vast majority of the bulk calcium measured with dendritic calcium imaging, the vigorous and brief calcium entry during Na-dSpikes is more critical for the induction of PP → CA1$_{tuft}$ LTP.

During TBS, Na-dSpikes produce the largest voltage excursions and maximize I$_{Ca}$ through both NMDAR and L-Ca$_v$ channels (**Figure 10**). Activation of NMDAR channels is strongly voltage dependent (through voltage-dependent magnesium block) as is activation of L-Ca$_v$ channels (through high-voltage-activated conformational changes). Thus, these calcium-permeable channels that we implicate in PP → CA1$_{tuft}$ LTP are activated more by the Na-dSpike than by the slow synaptic depolarization. The large I$_{Ca}$ caused by the Na-dSpike produces a brief, high calcium concentration near the mouth of the channels, which will quickly diffuse away and contribute modestly to the bulk calcium concentration in the rest of the dendritic spine and shaft. On the other hand, most of the bulk calcium is generated by the smaller, but longer-lasting flux of calcium through the NMDAR channels activated during the slow synaptic depolarization. The calcium concentration near the mouth of the channels is proportional to the instantaneous amplitude of I$_{Ca}$, while the bulk calcium concentration is proportional to the time integral of I$_{Ca}$ through the calcium-permeable channels (**Augustine et al., 2003**; **Eggermann et al., 2012**; **Tadross et al., 2013**).

The efficacy of low TTX as a blocker of LTP by inhibiting Na-dSpikes could be explained by the existence of a calcium sensor near the channels that triggers a series of LTP-inducing biochemical events (**Figure 10**). The affinity and co-operativity of the calcium sensor would have to be matched to

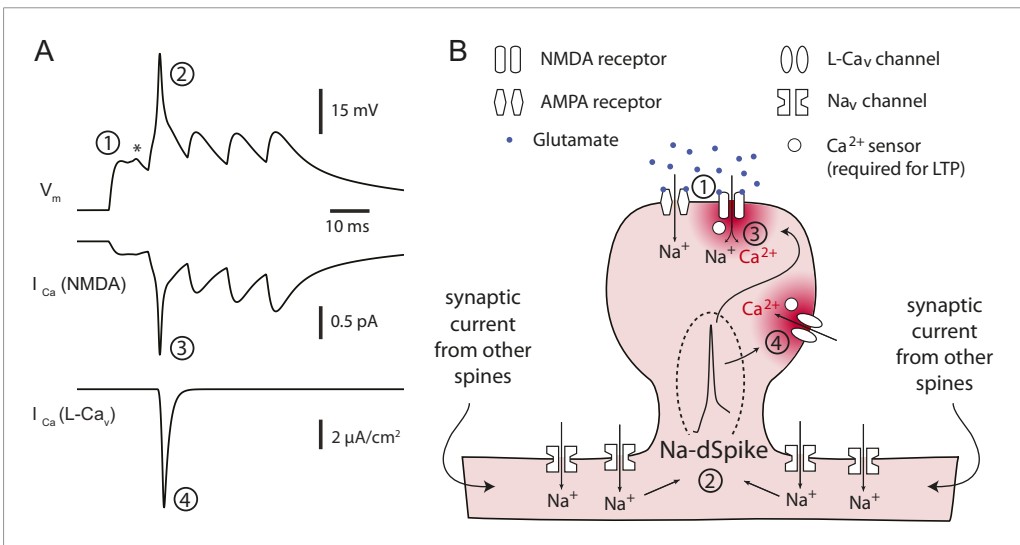

**Figure 10**. Proposed model for the critical role of Na-dSpikes in the induction of PP → CA1$_{tuft}$ LTP. (**A**) Synaptic membrane potential and calcium currents in response to a high-frequency burst activation of glutamatergic synapses (re-plotted from **Figure 8D**). (1) AMPA-/NMDA-EPSP, (2) Na-dSpike, (3) I$_{Ca}$ through NMDAR channels, (4) I$_{Ca}$ through L-Ca$_v$ channels. Note also the presence of a spikelet (asterisk). (**B**) Schematic illustration of the events leading to the induction of PP → CA1$_{tuft}$ LTP. Strong activation of PP → CA1$_{tuft}$ synapses results in EPSPs (1) and, on some trials, subsequently leads to initiation of Na-dSpikes (2). The locally generated Na-dSpike mediates the largest I$_{Ca}$ through both NMDAR channels (3) and L-Ca$_v$ channels (4), thus resulting in a high, localized calcium concentration near the mouth of the channels (dark red), which activates a series of biochemical events necessary for the induction of LTP. This intracellular calcium diffuses away, contributing only modestly to the 'bulk' calcium concentration throughout the dendritic spine and shaft (pink), which is eventually removed from the cytoplasm by pumps in the plasma membrane and in organelles such as the endoplasmic reticulum. In contrast, the smaller, longer-lasting I$_{Ca}$ through NMDAR channels generated during the slow synaptic depolarization produces a lower localized calcium concentration near the channel pore, but contributes more to the bulk calcium concentration.

the spatiotemporal features of the calcium concentration profiles near the mouths of NMDAR and L-Ca$_v$ channels in response to TBS. The affinity of the sensor would need to be relatively low, in order to respond *only* to the high calcium concentration near the calcium-conducting pores during a large Na-dSpike. As our model predicts a modest increase in peak I$_{Ca}$ (proportional to local calcium concentration) during a dSpike, relative to the slow synaptic depolarization, it is likely that the sensor would also need high cooperativity in order to respond selectively during the dSpike. Importantly, experimentally observed Hill coefficients are indeed high enough for this operation; for example, assuming a Hill coefficient of 4, a modest dSpike-driven difference in peak I$_{Ca}$ (~twofold; *Figure 8*) would increase calcium sensor activation from 20% to 80%. The identities of the calcium sensors and the downstream biochemistry leading to LTP induction are the subjects of ongoing research (*Amici et al., 2009*), but calmodulin may have the properties necessary to serve this function, namely low affinity, rapid binding kinetics, and high cooperativity for calcium binding (*Faas et al., 2011*). Calmodulin has been postulated to be the initial calcium sensor coupled to other key LTP-inducing molecules such as calcium/calmodulin-dependent protein kinase II (*Bradshaw et al., 2003*; *Lisman et al., 2012*).

Alternatively, Na-dSpikes might couple activation of NMDAR or L-Ca$_v$ channels to LTP via mechanisms that operate independently of postsynaptic calcium influx or in concert with it. For example, it is well known that the gating of L-Ca$_v$ channels is coupled to muscle contraction independently of calcium flux through the channels and that similar mechanisms may exist in neurons (*Kaczmarek, 2006*). Indeed, activation of NMDARs in the absence of calcium influx through the channels has been suggested to be sufficient for induction of long-term depression (LTD; *Nabavi et al., 2013*; but see *Babiec et al., 2014*). In keeping with this possibility, we observed LTD in neurons buffered with 10 mM BAPTA. However, the block of LTP by intracellular calcium buffering (*Figure 9*) suggests that NMDAR and L-Ca$_v$ channels contribute to the induction of PP → CA1$_{tuft}$ LTP by their ionotropic rather than metabotropic functions. Moreover, high-frequency stimulation of PP → CA1$_{tuft}$ synapses has been shown to result in LTP expression mediated by functional upregulation of presynaptic Ca$_v$ channels (*Ahmed and Siegelbaum, 2009*), but the lack of effect of nimodipine on synaptic responses after LTP induction (*Figure 4—figure supplement 1*) indicates that the effect of nimodipine on TBS-induced PP → CA1$_{tuft}$ LTP cannot be explained by the block of a presynaptic expression mechanism mediated by L-Ca$_v$ channels. Nevertheless, we cannot rule out a possible role of presynaptic NMDARs in LTP induction (*Kunz et al., 2013*).

The requirement for Na-dSpikes has interesting functional implications for PP → CA1$_{tuft}$ LTP. The activation of multiple synapses on a single dendritic branch leads to initiation of a dSpike (*Losonczy and Magee, 2006*), but dSpikes tend to fail at branch points (*Spruston, 2008*; *Katz et al., 2009*). Thus, the spatial extent of a Na-dSpike-dependent LTP event may be restricted to a particular dendritic branch, or even a segment of a branch containing the synapses that contribute to initiation of a Na-dSpike. If true, such compartmentalization of plasticity (*Losonczy et al., 2008*; *Makara et al., 2009*; *Makino and Malinow, 2011*) would have substantial implications for the granularity of associations learned at the dendritic level, and consequently, the information-carrying capacity of a pyramidal neuron, as information may be stored discretely in individual dendritic branches or segments, rather than in an entire neuron (*Poirazi and Mel, 2001*; *Mehta, 2004*; *Wu and Mel, 2009*).

## Comparison to other forms of LTP

There are many forms of LTP, so it is not clear how general the requirement for Na-dSpikes will prove to be, even at the same synapses but with a different induction protocol. PP → CA1$_{tuft}$ LTP has also been reported to occur in response to combined activation of the PP and the more proximal Schaffer collaterals (SCs; *Takahashi and Magee, 2009*). In those experiments, dendritic Na$_v$ channels would be critical because bAPs were necessary to trigger calcium plateau potentials (a form of Ca-/NMDA-dSpike) that induced LTP. For this reason, it may prove difficult to disentangle the contributions of dendritic regenerative events mediated primarily by dendritic Na$_v$ channels (such as Na-dSpikes and bAPs) from dendritic regenerative events mediated by other channels (such as Ca- and NMDA-dSpikes) to the induction of PP → CA1$_{tuft}$ LTP with this paradigm. Moreover, although NMDARs and R-type Ca$_v$ channels were shown to be important for this form of LTP (*Takahashi and Magee, 2009*), the role of dendritic L-Ca$_v$ channels is unclear. L-Ca$_v$ channels were not required for plateau potentials, but whether their activation by plateau potentials contributes to the induction of this form of PP → CA1$_{tuft}$ LTP has not been tested.

Given the remarkable capacity of the brain to store information, it is no surprise that there are many different forms of LTP. Thus, we expect our findings generalize to some synapses but not others. For example, LTP has been induced at SC synapses on CA1 pyramidal neurons using many different protocols, such as those that produce synaptic depolarization alone, without bAPs or dSpikes (*Gustafsson et al., 1987*; *Isaac et al., 1995*; *Liao et al., 1995*; *Han and Heinemann, 2013*). Another interesting form of LTP at SC synapses depends on an instructive signal from coincident PP activation (*Dudman et al., 2007*). This form of LTP does not depend on bAPs or dSpikes either, but is mediated by endocannabinoid release from dendrites, induced by summation of EPSPs from both pathways, which ultimately downregulates feedforward inhibition (*Xu et al., 2012*; *Basu et al., 2013*). At these same SC synapses, other forms of LTP have been reported to depend on postsynaptic action potential firing, presumably via bAPs (*Magee and Johnston, 1997*; *Thomas et al., 1998*; *Wittenberg and Wang, 2006*; *Buchanan and Mellor, 2007*). Still other forms of LTP at SC synapses, however, are correlated with the initiation of dSpikes (Remy and Spruston, 2007; *Hardie and Spruston, 2009*), but it is unclear what role Na-dSpikes play, relative to Ca-dSpikes or NMDA-dSpikes.

At excitatory synapses on other cell types, LTP has been reported to result from a range of postsynaptic mechanisms (*Gordon et al., 2006*; *Holthoff et al., 2006*; *Kampa et al., 2007*; *Froemke et al., 2010*; *Hsu et al., 2010*, *2012*; *Brandalise and Gerber, 2014*). Importantly, however, a recent study demonstrated that rhythmic whisker stimulation induced LTP in somatosensory cortex, even when the stimulus did not drive somatic action potential firing (*Gambino et al., 2014*). Thus, LTP can occur in vivo, in response to naturalistic simulation, even in the absence of axo-somatic spiking. Nevertheless, action potentials and bAPs are likely to be important for other forms of synaptic plasticity. The diversity of LTP induction mechanisms is likely to contribute to the powerful learning capacity of the brain.

## A potential role for Na-dSpikes in cognition at multiple timescales

In summary, the experiments presented here argue for a critical, causal role of dendritic $Na_v$ channels, by triggering Na-dSpikes, in the induction of PP → $CA1_{tuft}$ LTP. With this conclusion, we have now established possible functional roles for Na-dSpikes not only in the rapid, moment-to-moment processing of internal and external information (by influencing synaptic integration; *Golding and Spruston, 1998*; *Jarsky et al., 2005*; *Losonczy and Magee, 2006*; *Katz et al., 2009*), but also in the long-term storage of this information (by a Na-dSpike-dependent form of LTP). Dendritic $Na_v$ channels were previously known to contribute to the cellular basis of memory by mediating forms of LTP that require action potential backpropagation. In this case, the Hebbian plasticity rule—'neurons that fire together wire together'—is implemented by bAPs. Our results suggest that dendritic $Na_v$ channels can also contribute to memory formation via a critical role for Na-dSpikes in some forms of LTP. Our view is that this form of LTP is Hebbian on a finer spatial scale, for which the rule can be rephrased as 'neurites that fire together wire together'. Dendritic $Na_v$ channels play a central role in the cellular implementation of this rule, by determining which postsynaptic neurites fire.

## Materials and methods

### Slice preparation

All animal procedures were approved by the Animal Care and Use Committees at the HHMI Janelia Research Campus and Northwestern University. 3- to 7-week-old male Wistar rats were decapitated under deep isoflurane anesthesia, and the brain was transferred to an ice-cold dissection solution containing (in mM): 110 Choline-Cl, 0.2 NaCl, 2.5 KCl, 1.25 $NaH_2PO_4$, 25 $NaHCO_3$, 15 Dextrose, 2.4 Na-Pyruvate, 1.3 Na-Ascorbate, 0.5 $CaCl_2$, 3 $MgCl_2$ (pH 7.4, oxygenated with 95% $CO_2$ and 5% $O_2$). 300- to 350-μm thick, near-horizontal slices were sectioned using a vibrating tissue slicer (Vibratome 3000, The Vibratome Company, St. Louis, MO; or Leica VT 1200S, Leica Microsystems, Wetzlar, Germany). CA3 and superficial layers of the entorhinal cortex were removed to limit polysynaptic activation. The slices were transferred to a suspended mesh within a chamber filled with artificial cerebrospinal fluid (ACSF) containing (in mM): 119 NaCl, 2.5 KCl, 1.25 $NaH_2PO_4$, 25 $NaHCO_3$, 25 Dextrose, 2 $CaCl_2$, 1 $MgCl_2$ (3 Na-Pyruvate, 1 Na-Ascorbate were added when two-photon imaging experiments were performed; pH 7.4, oxygenated with 95% $CO_2$ and 5% $O_2$). After 30 min of incubation at 35°C, the chamber was maintained at room temperature.

## Whole-cell recording and stimulation

All recordings were performed using slices submerged in the recording chamber of an Axioskop 2 (Carl Zeiss Microscopy, Jena, Germany) or SliceScope (Scientifica, East Sussex, UK) upright microscope constantly perfused with oxygenated ACSF at 33–35°C. SR-95531 (2 μM) and CGP 52432 (1 μM) were added to block $GABA_A$ and $GABA_B$ receptors, respectively. Patch pipettes were pulled from thick-wall borosilicate glass and fire polished, resulting in resistance of 3–5 MΩ and 7–9 MΩ for somatic and dendritic recording, respectively, when filled with intracellular solution containing (in mM): 115 K-Gluconate, 20 KCl, 10 $Na_2$-phosphocreatin, 10 HEPES, 4 Mg-ATP, 0.3 Na-GTP, and 0.1% biocytin.

When EGTA or BAPTA was included in the intracellular solution, the basal level of cytoplasmic calcium concentration (∼50 nM for CA1 pyramidal neurons; *Maravall et al., 2000*; *Henneberger et al., 2010*) was maintained by adding $CaCl_2$ to the intracellular solution, calibrated based on the mass-action reactions between calcium ion and the particular calcium chelator used (calculated with MaxChelator by Chris Patton, http://web.stanford.edu/~cpatton/webmaxc/webmaxcS.htm). The dissociation constant ($K_d$) for $Ca^{2+}$ at 33°C with the ionic strength 0.16 N, pH 7.3 is 91.7 and 229 nM for EGTA and BAPTA, respectively. The concentrations of calcium chelator and $CaCl_2$ included are (in mM) 0.18 $CaCl_2$ for 0.5 EGTA; 3.53 $CaCl_2$ for 10 EGTA; 0.09 $CaCl_2$ for 0.5 BAPTA; 1.79 $CaCl_2$ for 10 BAPTA.

Dendritic recordings were obtained 200–320 μm away from the soma. Patch pipette series resistance was always lower than 50 MΩ. Recordings were made using a Dagan BVC-700A amplifier (Dagan Corporation, Minneapolis, MN). Data were low-pass filtered at 3 or 5 kHz and digitized at 50 kHz via an ITC18 digital-analog converter (HEKA Instruments Inc., Bellmore, NY) under control of custom macros programmed in IGOR Pro (Wavemetrics, Lake Oswego, OR).

For synaptic stimulation, theta-glass or capillary micropipettes (∼40–50 μm in diameter) filled with ACSF were used for bipolar or monopolar stimulation via a stimulus isolator (BSI-950, Dagan Corporation; or Model 4AD, Getting Instruments, San Diego, CA), placed in stratum lacunosum-moleculare (SLM) ∼200–300 μm away (usually toward subiculum) from the recorded neuron. For LTP induction, stimulus intensities were set to give EPSP amplitudes of 2–5 mV in somatic recordings and 4–10 mV in dendritic recordings, consistent with approximately 50% attenuation of EPSPs between the dendrite (200–300 μm from the soma) and the soma (*Golding et al., 2005*). For testing the effects of 20 nM TTX on synaptic transmission (*Figure 1*), a larger range of stimulus intensities (giving EPSP amplitudes of 3–10 mV in somatic recordings) was used. PP → CA1$_{tuft}$ EPSPs were monitored every 20 s, and interleaved test pulses were used to monitor the recording quality (series resistance, bridge balance, pipette capacitance compensation) and the input resistance of the cell throughout the experiment. To induce PP → CA1$_{tuft}$ LTP, we used TBS, which consisted of five burst stimuli grouped at 5 Hz, with each consisting of 5 synaptic stimuli at 100 Hz. This stimulus was repeated three times (TBSx3), at 30-s intervals. Each TBS was delivered under one of the following three conditions: (1) paired with brief (2 ms) somatic current injections at 50 Hz to evoke 3 action potentials during each burst (TBSx3+Current), (2) with the soma voltage-clamped at −68 to −70 mV (TBSx3+SomaticVC), or (3) alone (5-stim TBSx3). In most cases of somatic voltage clamp, the soma was clamped well enough to prevent action potential firing in response to TBS; however, in three cases, escape spikes were observed, so these experiments were rejected from the dataset.

For pharmacological experiments, PP → CA1$_{tuft}$ EPSP amplitude was monitored before and 10 min after bath application of drugs to confirm that synaptic transmission was unaffected. To block NMDARs, the D-isomer of AP5 was used to maximize the effect (*Watkins and Olverman, 1987*; *Morris, 1989*). Notably, 50 μM AP5 alone inhibited LTP as much as the combination of AP5 and MK-801 used in our previous work (*Golding et al., 2002*; Remy and Sprusten, 2007), suggesting that the limited block of LTP by AP5 could not be attributed to glutamate competing away binding of AP5 to NMDARs. When EGTA or BAPTA was included in the intracellular solution, cells were dialyzed for at least 20–25 min before the LTP induction protocol was applied.

For antidromic stimulation, bath application of CNQX (10 μM) and AP5 (50 μM) was used to block ionotropic glutamate receptors (in addition to the GABA receptor blockers included in all experiments), and a stimulating electrode (as described above) was placed in stratum oriens, ∼20–50 μm away from the recorded neuron. Several different stimulus intensities were used to trigger action potentials antidromically.

In some experiments (*Figure 1*), 10 μM TTX was applied manually via pressure through a patch pipette positioned near the soma. To limit TTX diffusion toward the apical dendrites, slices were

positioned to have bath flow directed from the apical dendrites toward the soma. During perisomatic TTX application, EPSP amplitude was monitored and remained unchanged. In one experiment, action potentials were not completely eliminated in response to burst stimulation, so it was excluded from the analysis of burst responses (but not from the analysis of single-shock EPSPs).

Nickel chloride ($Ni^{2+}$), SR-95531, 6-cyano-7-nitroquinoxaline-2,3-dione disodium salt hydrate (CNQX), biocytin, sodium pyruvate, (+)-sodium L-ascorbate, phosphocreatine disodium, adenosine 5′-triphosphate magnesium (Mg-ATP), guanosine 5′-triphosphate sodium (Na-GTP), 4-(2-hydroxyethyl) piperazine-1-ethanesulfonic acid (HEPES), potassium D-gluconate, dextrose, and choline chloride were from Sigma–Aldrich (St. Louis, MO). Calcium chloride and magnesium chloride were from Fluka (St. Louis, MO) or Sigma–Aldrich. D-(−)-2-amino-5-phosphonopentanoic acid (D-AP5), tetrodotoxin citrate (TTX), nimodipine, CGP 52432, EGTA, and BAPTA were from Tocris Bioscience (Minneapolis, MN). Sodium chloride, potassium chloride, sodium bicarbonate, and sodium phosphate monobasic were from Fisher Scientific (Waltham, MA).

## Two-photon calcium imaging

Calcium imaging in distal tuft dendrites of CA1 pyramidal neurons was performed using a galvanometer-based two-photon laser scanning system (Prairie Ultima; Prairie Technologies, Middleton, WI) equipped with an epifluorescence microscope (BX61WI; Olympus, Tokyo, Japan) and a water-immersion objective lens (40X, 0.8 NA; Olympus). Neurons were filled with 100 µM Oregon Green 488 BAPTA-1, hexapotassium salt (OGB-1; Invitrogen, Waltham, MA) and 50 µM Alexa Fluor 594 Hydrazide (AF-594; Invitrogen, Waltham, MA). An ultrafast, Ti:Sapphire pulsed laser (Chameleon Ultra; Coherent, Auburn, CA) was tuned to 880 nm to acquire reference images and 920 nm to perform calcium imaging. Laser power was controlled with an electro-optical modulator (Model 350-80; Conoptics, Danbury, CT). Line-scan imaging along dendritic shafts (typically 5–10 µm in length) and across spines was performed using the Ultima scanner at 250–500 Hz with a dwell time of 10 µs, and fluorescence was collected by multi-alkali photomultiplier tubes (PMTs; Hamamatsu Photonics, Hamamatsu City, Japan). Laser power was adjusted to ensure a good signal-to-noise ratio of the fluorescence signal without photo-bleaching the dyes or photo-damaging the dendrites (see below). In some experiments, multiple locations on several branches or longer stretches of the branch (up to ∼35 µm) were imaged to determine the spatial profile of the calcium signals. All the data acquisition and device controls were performed using BNC-2090 and BNC-2110 boards (National Instruments, Austin, TX) with Prairie View and TriggerSync software (Prairie Technologies).

To achieve an equilibrium concentration of the dyes sufficient for calcium imaging in the distal dendrites (distance from the soma: 312–673 µm; average = 411 ± 18 µm), cells were dialyzed for at least 30 min before imaging commenced. Calcium signals were quantified as the increase in (green) OGB-1 fluorescence from the baseline before stimulation (50–100 ms) divided by (red) AF-594 fluorescence ($\Delta G/R$). This quantification is insensitive to small variations in basal calcium concentration and independent of the volume of imaged structures (*Sabatini et al., 2002*). For calcium imaging, single high-frequency burst stimulation (instead of full TBS) was applied to the PP with an interval of at least 30–120 s; PP → $CA1_{tuft}$ EPSP amplitude was constantly monitored to determine the stability of synaptic responses.

Consistent with previous theoretical and experimental studies (*Helmchen et al., 1996*; *Sabatini et al., 2002*), control experiments showed that the integral of $\Delta G/R$ is a more stable measure of calcium entry than peak $\Delta G/R$ (*Figure 7—figure supplement 1*; *Figure 7—source data 2*). Control experiments suggested that dye saturation did not occur under our conditions (data not shown). The largest peak $\Delta G/R$ achieved in the distal dendrites (∼250%, by pairing TBS with somatic current injection to evoke bAPs) was well above the largest value observed in response to a single high-frequency burst stimulation. Furthermore, the dye was not saturated by a single burst in general because calcium responses were able to facilitate upon delivery of additional bursts in TBS with the ratio $\Delta G/R_{(Max)}:\Delta G/R_{(first\ burst)} = 1.52 \pm 0.31$ (n = 4). Importantly, no correlation was observed between normalized drug effects and the $\Delta G/R$ measure in control (data not shown), which would be expected if dye saturation had resulted in an underestimate of pharmacological inhibition. The following indications of phototoxicity were monitored carefully: basal fluorescence of both dyes and its ratio ($G_0/R$ as a readout of basal calcium concentration; see *Figure 7—figure supplement 1*), morphological changes of dendrites (swelling or fragmentation), long-lasting depolarization

following synaptic stimulation or current injection, and a sudden loss of calcium signals. Experiments were terminated if any signs of photo-damage were observed. At the end of each experiment, the path distance and the depth (usually 25–55 μm from the surface) of the imaging sites were measured, and high-resolution Z-stack images were collected.

In initial experiments, we compared the effects of 20 nM TTX across different PP → CA1$_{tuft}$ LTP induction protocols (see above), and found reduction of calcium signals in all conditions (data not shown). However, during high-frequency burst stimulation of the PP, bAPs could contribute to calcium signals in distal dendrites (data not shown). To focus on the calcium signaling mediated by dSpikes rather than bAPs (which are not required for the induction of PP → CA1$_{tuft}$ LTP; *Figure 3E*; *Golding et al., 2002*), we conducted calcium imaging with the soma voltage-clamped at ∼ −70 to −75 mV. No escape spikes were observed under these conditions in this series of experiments.

## Data analysis

For all recordings, the peak amplitude of events was measured as the difference between the resting and peak membrane potentials. For somatic recordings, the apparent voltage threshold of action potentials was measured as the voltage at which dV/dt crossed 40 V/s. In dendritic recordings, the voltage responses after each synaptic stimulus position (Stim. #) could be divided into two populations with non-overlapping amplitude distributions (*Figure 2—figure supplement 1A*). Large-amplitude dendritic events (defined as those with the peak amplitude >40 mV) were presumed to consist of both bAPs and large dSpikes. The apparent voltage threshold of large-amplitude dendritic events was measured as the voltage at which the second temporal derivative of voltage $d^2V/dt^2$ crossed 7.5 mV/ms$^2$. bAPs typically have a high apparent voltage threshold, because large dendritic EPSPs are required for the axonal EPSPs to be large enough to reach the threshold for action potential initiation after EPSP attenuation between the dendrites and the axon. In contrast, large dSpikes have a lower apparent threshold, because they are initiated closer to the dendritic recording electrode.

bAPs and large dSpikes also differ in their onset kinetics, quantified as 'initial phase slope', which was measured as the slope of a linear fit to the initial portion of the phase plot (dV/dt vs V; *Figure 2—figure supplement 2A*). As dV/dt is proportional to membrane current, the initial phase slope can be conceptualized as the *apparent* voltage sensitivity of membrane current. It is an *apparent* voltage sensitivity because the source of the current (e.g., the Na$_v$ channels) and the membrane potential responsible for the current are at a remote location, and therefore the observed voltage sensitivity does not necessarily reflect the voltage dependence of the activation of the channels per se. In dendritic recordings, actively propagating spikes initiated in the axon (i.e., bAPs) typically have a sharp 'kink' at their onset, because they are initiated by axial current flowing from a remote location of spike initiation (i.e., the axon), which precedes local dendritic Na$_v$ channel-mediated currents. The 'kink' is reflected as a large initial phase slope (>3.5 ms$^{-1}$), and thus a steep apparent voltage sensitivity of the membrane current, because the axial current is driven by the voltage difference between the axon and the dendrite, thus resulting in a current that is not driven by changes in the local dendritic depolarization and therefore does not require activation of local channels. As a result of this sudden current, the onset of these backpropagating spikes shows a kink in the time domain and a steep initial slope in the phase domain. In contrast, large dSpikes have a more gradual onset, because they are initiated by local Na$_v$ channel-mediated currents generated closer to the dendritic recording electrode, and therefore the voltage sensitivity of the membrane current (as measured by initial phase slope) more accurately reflects the voltage sensitivity of the Na$_v$ channels (*Shu et al., 2007*; *Yu et al., 2008*; *Brette, 2013*; *Smith et al., 2013*).

On the basis of these two criteria, some events could be identified as clear bAPs (high threshold and large initial phase slope) or clear large dSpikes (low threshold and small initial phase slope), while the other events were ambiguous (*Figure 2—figure supplement 2*; *Figure 2—source data 2*), perhaps owing to complications such as coincident local dendritic depolarization (which may reduce the initial phase slope of bAPs) or very distal locations of dSpike initiation (which may result in large initial phase slope of propagating dSpikes). Three large-amplitude dendritic events with relatively small initial phase slope but high threshold also had relatively broad halfwidth, suggesting a larger contribution from Ca$_v$ and/or NMDAR channels (*Figure 2—figure supplement 2B*; *Figure 2—source data 2*).

Small dSpikes (spikelets) were identified as outliers in the distribution of peak dV/dt values (stimulus positions with a large-amplitude dendritic event in the control condition were excluded for

this analysis) for each of the five stimulus positions (Stim. #) in each burst (*Figure 2C*). We observed a trend toward decay of peak dV/dt in each burst as a function of stimulus position (*Figure 2—figure supplement 1B*). Therefore, we normalized peak dV/dt to the median value for Stim. #1 in control (i.e., of the five bursts in control from the given cell) and plotted normalized dV/dt as a function of stimulus position in each burst in control and in the presence of 20 nM TTX. A fit of an exponential decay function to the whole population of data (control and 20 nM TTX) was then performed. Small dSpikes were identified as events that fell outside the prediction band of the fit; in this way, a single criterion was applied to both experimental conditions (i.e., control and 20 nM TTX). Because there is no perfectly objective way to determine which events should be called small dSpikes, we analyzed the data for prediction bands at confidence levels ranging from 85% (least stringent; largest number of small dSpikes) to 99% (most stringent; smallest number of small dSpikes). Two additional statistical criteria (based on the median or standard deviation of normalized peak dV/dt for each stimulus position; data not shown) and another approach applying a single 'hard' threshold (*Figure 2—figure supplement 4*) for identifying small dSpikes were also used, and the qualitative results remained the same.

For notched box plots (*Figure 2—figure supplement 4A*), the edges of notch were defined by the 95% confidence intervals of the median, the end points of whiskers as the largest and smallest values that are within the range of (the 75th percentile + 1.5 × interquartile range) and (the 25th percentile − 1.5 × interquartile range), and individual data points are those outside this range.

The potentiation ratio was calculated as the average EPSP amplitude 26–30 min after LTP induction normalized by the average EPSP amplitude before induction.

Calcium and current signals were averaged over 3–6 trials, with a slight Gaussian smoothing (standard deviation of the kernel = 0.064 ms), and the peak amplitude and integral (up to 800 ms after stimulation) were measured. Special caution was taken to confirm that the smoothing reduced random noise without changing the kinetics and amplitude of the original signals. Tissue autofluorescence was determined to be negligible in the two-photon experiments, so no background subtraction was carried out. The two-photon Z-stack images of neurons were generated by two-dimensional projection of maximal AF-594 fluorescence intensity from three-dimensional Z-stacks.

All analyses were performed using IGOR Pro (Wavemetrics, Lake Oswego, OR), MATLAB (The MathWorks, Natick, MA), ImageJ (National Institutes of Health, Bethesda, MD), and Microsoft Office Excel (Microsoft, Redmond, WA).

In most cases (unless noted otherwise), data were presented as mean ± S.E.M. Significance of the difference in normalized EPSP amplitude between experimental conditions was tested by one-way ANOVA with *post hoc* multiple comparisons by Tukey's method. Significance of LTP expression for each experimental condition was tested by one-way repeated measures ANOVA. Significance of the difference in number of events counted as small dSpikes (spikelets) between two experimental conditions was tested using a binomial test; rejection of the null hypothesis suggests an unequal likelihood of an event occurring above threshold between two conditions. All other comparisons between two experimental groups/conditions were done with Student's t-test. Statistical analyses were performed using Prism (GraphPad Software, La Jolla, CA) and OriginPro (OriginLab Corporation, Northampton, MA). All data presented in the figures has been deposited to our laboratory website (www.janelia.org/lab/spruston-lab/resources) and figshare (*Spruston et al., 2015*).

## Computational modeling

Simulations were performed using a compartmental model based on the morphology reconstructed from a rat CA1 pyramidal neuron (*Golding et al., 2001*). The passive and active properties of the model were used the same as those used previously, which reproduced experimental data on bAPs and dSpikes (*Golding et al., 2001*; *Jarsky et al., 2005*; *Katz et al., 2009*) with adaptations described below. All simulations were performed using the NEURON simulation software (*Hines and Carnevale, 1997*) with a fixed time step (*dt* = 0.025 ms). Code for the simulations has been deposited to our laboratory website (www.janelia.org/lab/spruston-lab/resources) and the ModelDB database (*Hsu et al., 2015*).

The model included active conductances simulating the following channels: $Na_v$ channels, A-type potassium ($K_A$) channels, delayed-rectifier potassium ($K_{DR}$) channels, and L-$Ca_v$ channels. Properties and distributions of $K_A$ and $K_{DR}$ channels were the same as previously stated (*Golding et al., 2001*; *Katz et al., 2009*). The L-$Ca_v$ channel model (*Poirazi et al., 2003*) was inserted into the apical tuft with

a uniform density $g_{Ca}$ = 1.25 mS/cm². The properties of the dendritic $Na_v$ channels, including slow inactivation, were implemented using a model from *Migliore et al., 1999*. Inclusion of slow inactivation improved the fit of the limited effect of 20 nM TTX on the dendritically recorded voltage in response to single high-frequency burst stimulation (*Figures 2A,B, 5*). Following a recent study (*Lorincz and Nusser, 2010*), in the apical dendritic tree a linearly decreasing gradient of $Na_v$ channel density was implemented using the equation:

$$g_{Na}(x) = g_{Na,soma} + \Delta g_{Na} \cdot x$$

where $x$ is the distance from the soma (in µm), and $g_{Na}(x)$ is the $Na_v$ channel conductance at the distance $x$; $g_{Na,soma}$ = 42 mS/cm², $\Delta g_{Na} = -0.025$ (mS/cm²)/µm, and for the remaining compartments, $g_{Na} = g_{Na,soma}$.

Synaptic stimulation was simulated as 150 simultaneous events distributed randomly across all of the apical tuft branches. Spines were not explicitly modeled, but the associated surface area was accounted for by decreasing specific membrane resistivity ($R_m$) and increasing specific membrane capacitance ($C_m$) by a factor of 2 for compartments >100 µm from the soma. In additional simulations, synapses simulated on explicitly modeled spines yielded qualitatively similar results. Every synapse was taken to have α-amino-3-hydroxy-5-methyl-4-isoxazolepropionic acid (AMPA) receptor (AMPAR) and NMDAR conductances ($g_{AMPA}$ = 0.18 nS, $g_{NMDA}$ = 0.18 nS), and both were modeled as a difference of two exponentially decaying functions with rise and decay time constants of 0.2 and 2 ms for AMPARs (*Katz et al., 2009*) and 1 and 50 ms for NMDARs (*Hestrin et al., 1990*; *Spruston et al., 1995*). The voltage-dependent magnesium ($Mg^{2+}$) block of NMDARs was modeled as $g_{Mg} = [1 + 0.2801 \cdot Mg_{ext}^{2+} \cdot exp(-0.062 \cdot (V - 10))]^{-1}$ (*Spruston et al., 1995*; *Maex and De Schutter, 1998*) with the $Mg^{2+}$ concentration in bath $Mg_{ext}^{2+}$ = 1 mM. The fractional calcium influx through NMDAR channels was taken to be 10% of the total current (*Schneggenburger et al., 1993*; *Burnashev et al., 1995*; *Garaschuk et al., 1996*). The temperature used in the model was 35°C.

A calcium handling mechanism was implemented (*Hines and Carnevale, 2000*) which included buffers (endogenous buffer and OGB-1), diffusion (radial and longitudinal), and extrusion (a pump). The parameters of OGB-1 were as follows (*Schmidt et al., 2003*): diffusional mobility 0.3 µm²/ms, calcium-binding forward rate constant 430 mM⁻¹ms⁻¹, backward rate constant 0.14 ms⁻¹, and effective concentration 0.05 mM. The endogenous buffer was immobile, and the parameters were as follows (*Yamada et al., 1989*): calcium-binding forward rate constant 100 mM⁻¹ms⁻¹, backward rate constant 0.1 ms⁻¹, and concentration 0.003 mM. The calcium extrusion by a calcium pump was modeled as a two-step reaction with mass-action kinetics:

$$Ca_{in}^{2+} + pump \rightleftarrows Ca^{2+} \times pump$$

$$Ca^{2+} \times pump \rightleftarrows pump + Ca_{out}^{2+}$$

The forward and backward rate constants for the first step were 1 mM⁻¹ms⁻¹ and 0.005 ms⁻¹, and for the second step 1 ms⁻¹ and 0.005 mM⁻¹ms⁻¹, respectively. The calcium pump density was $10^{-11}$ mol/cm². With these calcium buffers and calcium pump, the free calcium concentration ($[Ca^{2+}]_{free}$) was in the linear operation range of OGB-1 (i.e., the dissociation constant), and the kinetics of $[Ca^{2+}]_{OGB}$ signals matched the imaging data.

To simulate bath application of 20 nM TTX, we reduced the conductance of $Na_v$ channels to 50% of control, consistent with the previous observations of TTX efficacy on hippocampal neurons (*Kaneda et al., 1989*; *Madeja, 2000*); to simulate bath application of 50 µM AP5 or 10 µM nimodipine, we reduced the NMDAR or L-$Ca_v$ channel conductance to zero. For all simulations, either a somatic voltage clamp at −70 mV was simulated or $g_{Na,soma}$ and $g_{Na,axon}$ were set to be zero in order to mimic the experimental paradigm in which axo-somatic action potential firing was prevented.

Although the model contains a large number of parameters, many of these values were constrained by a wealth of available experimental data on CA1 pyramidal neurons (this study; *Magee and Johnston, 1995*; *Spruston et al., 1995*; *Helmchen et al., 1996*; *Hoffman et al., 1997*; *Golding et al., 2001*; *Otmakhova et al., 2002*; *Sabatini et al., 2002*; *Katz at al., 2009*; *Bittner et al., 2012*). As in all modeling, some assumptions had to be made, so we were careful to test a range of parameter values. Our conclusions were robust using several combinations of parameters. As always, however, we make no claim that the model is 'correct' in an absolute sense. Its purpose is to demonstrate the plausibility

of our interpretation and the proposed model with parameter values constrained by those available in the literature.

## Acknowledgements

We thank Mike Tadross and members of the Spruston lab for helpful discussions and Aaron Milstein, Samuel Gale, and Mark Harnett for advice on programming and experiments.

## Additional information

### Funding

| Funder | Grant reference | Author |
| --- | --- | --- |
| Howard Hughes Medical Institute (HHMI) | | Yujin Kim, Ching-Lung Hsu, Mark S Cembrowski, Brett D Mensh, Nelson Spruston |
| National Institutes of Health (NIH) | NS035180 | Nelson Spruston |
| National Institutes of Health (NIH) | NS046064 | Nelson Spruston |
| National Science Council Taiwan | NSC98-2917-I-002-145 | Ching-Lung Hsu |

The funders had no role in study design, data collection and interpretation, or the decision to submit the work for publication.

### Author contributions

YK, C-LH, Conception and design, Acquisition of data, Analysis and interpretation of data, Drafting or revising the article; MSC, Acquisition of data, Analysis and interpretation of data, Drafting or revising the article; BDM, Analysis and interpretation of data, Drafting or revising the article; NS, Conception and design, Analysis and interpretation of data, Drafting or revising the article

### Ethics

Animal experimentation: All animal procedures were approved by the Animal Care and Use Committees at the HHMI Janelia Research Campus and Northwestern University.

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
