## [Decision Letter]

Thank you for sending your work entitled “Dendritic sodium spikes trigger long-term potentiation at distal synapses on hippocampal pyramidal neurons” for consideration at *eLife.* Your article has been favourably evaluated by Eve Marder (Senior Editor), Gary Westbrook (Reviewing Editor), and two reviewers. The Reviewing Editor and the other reviewers discussed their comments before reaching this decision, and the Reviewing Editor has assembled the following comments to help you prepare a revised submission.

This study characterizes the effects of low concentrations of TTX on the generation of dendritic action potentials and the induction of LTP in response to theta burst stimulation of perforant path (PP) synaptic inputs to distal dendrites of CA1 pyramidal neurons. Earlier work from this lab demonstrated the remarkable electrical isolation of distal dendrites from the rest of CA1 pyramidal cells, in which local spikes play an important role in local computations, and are also important in triggering LTP. The background for this study is the nature of these dendritic spikes. Dendrites are capable of a number of regenerative processes, including sodium spikes, high threshold calcium spikes and NMDAR-dependent “spikes”. The authors conclude that 20 nM TTX selectively inhibits dendritic spikes without any apparent effect on the amplitude of subthreshold synaptic responses. The low dose of TTX is surprisingly effective in blocking theta burst LTP, which is dependent on calcium influx through NMDA receptors and L-type voltage gated calcium channels. The authors present evidence that the sodium spikes engage L-type calcium channels as well as unblock NMDARs, both of which contribute to LTP.

Both reviewers thought that the topic was interesting and that the experiments were well-performed, however the reviewers raised important concerns regarding whether the results prove that Na-dependent dendritic spikes “trigger” or “cause” or whether they are contributors to the observed plasticity along with L-type calcium channels and NMDA receptors. *eLife* policy is not to extend the review process by demanding major sets of additional experiments that would require months and/or detailed re-review. Thus as you review the comments below, please consider which experiments are essential to prove that dendritic Na spikes “contribute” and which are necessary to support the stronger conclusion that dendritic sodium spikes are causal and “trigger” plasticity as the title implies. The revised Discussion and conclusions should match the experiments in that regard. We regard the experiments suggested in point 2 of reviewer 1 and point 3 of reviewer 2 to be essential.

*Reviewer 1*:

1) Reviewer 1 questioned the strength of the conclusion that block of dendritic spikes by TTX is causally related to the effect of TTX to block LTP. This conclusion is based on a single concentration of TTX, and the argument that it is unlikely that the voltage-gated sodium channels are acting through means other than dendritic spikes. However, Na_v_ channels can contribute to synaptic activation and influence calcium levels in individual spines (Bloodgood and Sabatini, Neuron 2007; Araya et al., PNAS, 2007) and conceivably could influence calcium indirectly by Na elevation and a decreased activation of Na/Ca exchange (e.g. Huang and Trussell, Neuron 2014). Can the authors demonstrate a more convincing causal link between the Na spikes and LTP? For example, can block of LTP be rescued by direct injection of current to simulate dendritic spikes, or do changes in spikes by a wider range of low TTX concentrations correlate with changes in LTP (i.e. do the two have similar IC50s?).

2) The authors fail to address certain discrepancies between their results and previous studies of synaptically evoked calcium signals in CA1 distal dendrites. The authors conclude that the L-type calcium channels participate in LTP by contributing to postsynaptic calcium influx. However, two previous studies using related stimulation protocols failed to observe any effect of L-type channel antagonists on synaptically evoked calcium signals in distal dendrites (Takahashi and Magee, Neuron 2009; Tsay et al., Neuron 2007). Did the authors examine the effects of L-type blockers on calcium levels under the conditions of their calcium imaging experiments?

One possible explanation for the discrepancy between effects of L-type blockers on postsynaptic calcium versus LTP is that the L-type channels are acting presynaptically to enhance transmitter release, consistent with a previous study (that the authors fail to discuss) demonstrating that PP LTP can have a significant presynaptic component of expression (Ahmed and Siegelbaum, Neuron 2009).

3) Figure 1. Given that local synaptic responses in the distal dendrites and isolated spines are likely to be significantly larger than the 3-10 mV depolarization in the soma, and thus capable of activating spine or shaft voltage-gated sodium channels, it is surprising that 20 nM TTX had no effect on the subthreshold synaptic response. Is it possible that the 10 μM local TTX applied to the soma had some effect distally? Did the authors try applying 20 nM TTX in the absence of local somatic TTX (using weak subthreshold distal stimulation)?

4) Figures 7, 8 and 9. To explain the discrepancy in their results in which Na_v_ channel blockade causes only a small decrease in total calcium signal yet fully blocks LTP, the authors postulate that LTP depends on a local calcium microdomain located near the inner mouth of L-type calcium channels. The authors could test this interesting idea by showing that LTP is relatively insensitive to block with the slow calcium buffer EGTA but is efficiently blocked by the fast buffer BAPTA (in contrast to previous studies at Schaffer collateral synapses showing LTP is blocked efficiently by EGTA).

Minor comment

1. Why do the authors use such a high affinity calcium dye (OGB-1)? Previous studies on dendritic calcium signals generally use much lower affinity dyes to prevent saturation. If OGB-1 is saturated by high dendritic calcium levels, this would lead the authors to underestimate the effects of the pharmacological inhibitors on calcium levels.

Reviewer 2:

The experiments are well crafted and the conclusions are generally reasonable. I have only a few comments.

1) Some of the figures are difficult to read, e.g., Figure 2. How do the conditions differ between the two voltage traces?

2) If the somatic APs are not necessary for LTP (voltage clamp has not impact), then why even do the experiment with the APs?

3) The Ca signal in the shaft that is blocked by AP5 is not necessarily coming directly of NMDARs in the spines. It could also involve the NMDAR-dependent depolarization of the shaft and activation of Ca_v_ channels on the shaft. This could be addressed by looking at nimodine alone and with TTX.

4) This is confusing. Na-dSpikes “reducing calcium influx? Just the opposite.

5) The Discussion is very lengthy and could easily be reduced by at least a third.

---

## [Author Response]

*Both reviewers thought that the topic was interesting and that the experiments were well-performed, however the reviewers raised important concerns regarding whether the results prove that Na-dependent dendritic spikes “trigger” or “cause” or whether they are contributors to the observed plasticity along with L-type calcium channels and NMDA receptors. eLife policy is not to extend the review process by demanding major sets of additional experiments that would require months and/or detailed re-review. Thus as you review the comments below, please consider which experiments are essential to prove that dendritic Na spikes “contribute” and which are necessary to support the stronger conclusion that dendritic sodium spikes are causal and “trigger” plasticity as the title implies. The revised Discussion and conclusions should match the experiments in that regard. We regard the experiments suggested in point 2 of reviewer 1 and point 3 of reviewer 2 to be essential*.

We understand why the word “trigger” may have been too strong. We meant it as a synonym for “necessary” or “necessary step” but it could be interpreted to mean that they are sufficient. Thus, we have changed the title to say that dendritic sodium spikes are “required” for LTP at PP-CA1_tuft_ synapses.

Reviewer 1:

*1) Reviewer 1 questioned the strength of the conclusion that block of dendritic spikes by TTX is causally related to the effect of TTX to block LTP. This conclusion is based on a single concentration of TTX, and the argument that it is unlikely that the voltage-gated sodium channels are acting through means other than dendritic spikes. However, Na*_*v*_
*channels can contribute to synaptic activation and influence calcium levels in individual spines (Bloodgood and Sabatini, Neuron 2007; Araya et al., PNAS, 2007) and conceivably could influence calcium indirectly by Na elevation and a decreased activation of Na/Ca exchange (e.g. Huang and Trussell, Neuron 2014). Can the authors demonstrate a more convincing causal link between the Na spikes and LTP? For example, can block of LTP be rescued by direct injection of current to simulate dendritic spikes, or do changes in spikes by a wider range of low TTX concentrations correlate with changes in LTP (i.e. do the two have similar IC50s?)*.

This is a reasonable point, and one that we cannot completely rule out. It is possible that subthreshold depolarization contributed by Na_v_ channels (which is less likely according to our data) or Na^+^ entry into spines is somehow critical for the induction of LTP, independently of Na-dSpikes. We have added text to the Discussion to address these possibilities. However, we feel that the data and modeling we present, including the new experiments (see below), make a strong case for the importance of Na-dSpikes driving the critical calcium influx at the mouth of L-Ca_v_ channels and NMDAR channels. We hope the reviewers and editors agree that discussing alternatives is an acceptable alternative to absolute proof, which is hard (maybe even impossible) to achieve.

We note also that we used a single concentration of TTX because there is a narrow range over which this experiment is likely to work. Higher concentrations of TTX would block synaptic transmission and lower concentrations would be expected to have a smaller effect on LTP and Na-dSpikes. Assaying the concentration dependence of these effects would consume a lot of time and resources; it is not clear to us what the benefit of that effort would be.

We opted not to do the experiments involving direct current injection into dendrites for a couple of reasons. First, the experiments are very difficult, as they involve maintaining dendritic recordings long enough to induce and monitor LTP. Second, even if we were to perform these experiments, a negative result would be difficult to interpret, because we would not know if we were able to induce the right amount of depolarization at the right time and in the right place in the vast expanse of distal dendrites.

2) The authors fail to address certain discrepancies between their results and previous studies of synaptically evoked calcium signals in CA1 distal dendrites. The authors conclude that the L-type calcium channels participate in LTP by contributing to postsynaptic calcium influx. However, two previous studies using related stimulation protocols failed to observe any effect of L-type channel antagonists on synaptically evoked calcium signals in distal dendrites (Takahashi and Magee, Neuron 2009; Tsay et al., Neuron 2007). Did the authors examine the effects of L-type blockers on calcium levels under the conditions of their calcium imaging experiments?

One possible explanation for the discrepancy between effects of L-type blockers on postsynaptic calcium versus LTP is that the L-type channels are acting presynaptically to enhance transmitter release, consistent with a previous study (that the authors fail to discuss) demonstrating that PP LTP can have a significant presynaptic component of expression (Ahmed and Siegelbaum, Neuron 2009).

We agree that the studies cited by the reviewer reported no effects of L-Ca_v_ channel blockers on dendritic calcium signals, but we do not view this as a discrepancy. Rather, we think the relevant localized calcium elevations near L-Ca_v_ channels do not contribute substantially to the “bulk” calcium concentration measured by calcium imaging in dendrites. To test this, we conducted additional calcium imaging experiments and we indeed found that 10 µM nimodipine had little effect on dendritic calcium signals during high-frequency burst stimulation (new figure: Figure 8—figure supplement 1). This is consistent with our model, presented in the Discussion that L-Ca_v_ channel-mediated calcium influx contributes to TBS-induced PP-CA1_tuft_ LTP by activating plasticity-related downstream signaling near the mouth of the channels, without providing a large contribution to the overall dendritic calcium influx during TBS. In addition, we experimentally ruled out the possibility that nimodipine blocks LTP by masking presynaptic contribution of L-Ca_v_ channels to LTP expression (Figure 4—figure supplement 1). Finally, our hypothesis is now directly supported by additional new experiments (Figure 9; see below).

*3)*
Figure 1*. Given that local synaptic responses in the distal dendrites and isolated spines are likely to be significantly larger than the 3-10 mV depolarization in the soma, and thus capable of activating spine or shaft voltage-gated sodium channels, it is surprising that 20 nM TTX had no effect on the subthreshold synaptic response. Is it possible that the 10 μM local TTX applied to the soma had some effect distally? Did the authors try applying 20 nM TTX in the absence of local somatic TTX (using weak subthreshold distal stimulation)?*

Yes, we have applied 20 nM TTX without applying 10 µM TTX perisomatically. These data were included in the Results section (control: 4.57 ± 0.36 mV; TTX: 5.02 ± 0.49 mV, n = 7 cells), before we introduced the concept of applying 10 µM TTX at the soma, which was done to prevent spiking when bursts of stimuli were applied.

One possible reason that 20 nM TTX had marginal or no effect on subthreshold synaptic responses is that this low concentration of TTX blocks only some of the Nav channels. However, [12] also showed that 1 µM TTX did not affect EPSPs from proximal dendritic spines of CA1 pyramidal neurons.

To provide further clarification, the only experiments where we used 10 µM TTX perisomatically are shown in Figure 1. In the LTP experiments, where 20 nM TTX was an effective blocker of LTP, 10 µM TTX was not applied perisomatically. Instead, we either allowed somatic/axonal spikes to occur during LTP induction or we applied somatic voltage clamp to prevent spiking.

*4)*
Figures 7, 8 and 9*. To explain the discrepancy in their results in which Na channel blockade causes only a small decrease in total calcium signal yet fully blocks LTP, the authors postulate that LTP depends on a local calcium microdomain located near the inner mouth of L-type calcium channels. The authors could test this interesting idea by showing that LTP is relatively insensitive to block with the slow calcium buffer EGTA but is efficiently blocked by the fast buffer BAPTA (in contrast to previous studies at Schaffer collateral synapses showing LTP is blocked efficiently by EGTA)*.

To address this point, we performed additional LTP experiments under four different conditions of intracellular calcium buffering: 0.5 mM EGTA, 10 mM EGTA, 0.5 mM BAPTA, and 10 mM BAPTA (in all cases the intracellular equilibrium basal calcium concentration was adjusted to 50 nM). As we predicted, only 10 mM BAPTA was able to block LTP. The new data are presented in Figure 9. We hope the reviewer will agree that this strengthens the case for localized calcium entry activating plasticity-related downstream signaling responsible for LTP, and also supports our model which is based on not only activation of dendritic Na_v_ channels, but also Na-dSpikes.

Minor comment

1. Why do the authors use such a high affinity calcium dye (OGB-1)? Previous studies on dendritic calcium signals generally use much lower affinity dyes to prevent saturation. If OGB-1 is saturated by high dendritic calcium levels, this would lead the authors to underestimate the effects of the pharmacological inhibitors on calcium levels.

We used OGB-1 in order to facilitate identification of dendritic calcium signals that were much harder to detect with lower-affinity dyes OGB-5N and OGB-6F (data not shown). A major difference between our and previous studies (such as [122] and [119]) is that we used a milder stimulation paradigm, which did not elicit dendritic plateau potentials. Thus, we needed the high-affinity dye to observe dendritic calcium signals reliably. We did two things to address the reviewer’s concern about dye saturation: first, to see the largest calcium responses achievable by physiological activities in our settings, we performed experiments assaying the relationships between ΔG/R and various patterns of synaptic stimulation (with 1.3 or 2 mM Ca2+) or different numbers and frequencies of bAPs evoked bysomatic current injection (because bAPs have considerable activity-dependent attenuation along the dendrites, the latter had to be performed in more proximal dendrites, ~15-90 μm from the soma). We found the largest possible peak ΔG/R values (observed when TBS was paired with somatic current injection to evoke bAPs) were well above the largest ΔG/R in response to single high-frequency burst stimulation in our dataset, and the dye was not saturated by a single burst in general because calcium responses were able to facilitate upon delivery of additional bursts in TBS (five bursts) with the ratio ΔG/R _(Max)_: ΔG/R _(1st burst)_ = 1.52 ± 0.31 (n = 4 cells, in 2 mM Ca^2+^) (Figure 11). In addition, we plotted our pharmacological data to determine if there was any positive correlation between drug effect and the ΔG/R in control, which would be expected if dye saturation had led to significant underestimate of pharmacological inhibition for larger responses; however, no such relationship was observed (Figure 12). These new data are provided here for the reviewer, but we decided not to clutter the paper with additional supplements. Instead, we added a note to the Materials and methods section.

Author response image 1.Dye saturation does not occur under our experimental conditions. *Top*. Summary of peak ∆G/R from dendritic shaft in response to different stimulation patterns. Only the experiments in which TBS of the PP paired with somatic current injection to evoke bAPs are shown here to demonstrate that the largest peak ΔG/R achievable by physiological activities in our settings was above the largest ΔG/R in response to single high‐frequency burst stimulation in the dataset (Figure 7 and Figure 8—figure supplement 1). A total of 15 different stimulation conditions (i.e., combinations of different calcium concentrations in the ACSF, synaptic stimuli and somatic current injections) were tested. *Bottom*. Representative traces of ∆G/R from dendritic shaft and voltage from somatic current-clamp recording in response to TBS paired or not paired with somatic current injection. Also note that the dye was not saturated by a single burst in general because ΔG/R was able to facilitate upon delivery of additional bursts in TBS with the ratio ΔG/R_(Max)_:ΔG/R_(1st burst)_ = 1.52 ± 0.31 (n = 4 cells, in 2 mM Ca^2+^).**DOI:**
http://dx.doi.org/10.7554/eLife.06414.037

Author response image 2.Relationship between dendritic calcium responses in control condition and normalized drug effects on the responses. Summary of effects of different pharmacological agents on ∆G/R (normalized to control) from dendritic shaft in response to high‐frequency burst stimulation plotted as a function of the ∆G/R in control condition. The lack of correlation between the magnitude of the drug effect and the initial response in control suggests that dye saturation does not affect our conclusions with respect to the drug effects.**DOI:**
http://dx.doi.org/10.7554/eLife.06414.038

Reviewer 2:

*1) Some of the figures are difficult to read, e.g.,*
Figure 2*. How do the conditions differ between the two voltage traces?*

In Figure 2, the two pairs of traces are examples from two different neurons and the two different colors correspond to control and low TTX conditions. We are not sure why this is unclear, as these details are provided in the legend and on the figure itself. Could it be due to a low resolution version of the figure used for the review process?

2) If the somatic APs are not necessary for LTP (voltage clamp has not impact), then why even do the experiment with the APs?

We did the experiments without voltage clamp first. We feel that this provides important information, mimicking *in vivo* conditions where CA1 cells are sometimes firing in response to synaptic input to the tuft (presumably owing to coincident activation of the Schaffer collaterals from CA3). Had we performed all of the experiments under voltage clamp (i.e. without AP firing) we would likely have been criticized for not doing experiments under more natural conditions, where backpropagating action potentials might have contributed to TBS-induced PP-CA1_tuft_ LTP as well.

*3) The Ca signal in the shaft that is blocked by AP5 is not necessarily coming directly of NMDARs in the spines. It could also involve the NMDAR-dependent depolarization of the shaft and activation of Ca*_*v*_
*channels on the shaft. This could be addressed by looking at nimodine alone and with TTX.*

We performed additional experiments and found that nimodipine had almost no measurable effect on dendritic calcium influx (Figure 8—figure supplement 1). Nevertheless, the reviewer is correct that Ca_v_ channels (other than L-type) could contribute to some of the AP5-sensitive responses. However, the lack of effect of Ni^2+^ on LTP (Figure 4) makes it seem unlikely that a component of calcium influx attributable to T-type or R-type Ca_v_ channels contributes to LTP. The contribution of N-type or P/Q-type channels is difficult to address, owing to the critical roles of these channels in presynaptic release.

*4) This is confusing. Na-dSpikes “reducing calcium influx? Just the opposite*.

Yes, we apologize. This error has been corrected.

*5) The Discussion is very lengthy and could easily be reduced by at least a third*.

We have done our best to reduce the length of the Discussion while still addressing the points raised above. For example, we removed two paragraphs about different forms of LTP induced by NMDARs versus L-Ca_v_ channels.